# FOREGROUND OR BACKGROUND? VISUAL INTERPRETABILITY AND ROBUSTNESS ANALYSIS OF CLIP

## ABSTRACT

Contrastive vision–language models (VLMs) such as CLIP achieve strong zero-shot recognition yet remain vulnerable to spurious correlations, particularly background over-reliance. We introduce Cluster-based Concept Importance (CCI), a novel interpretability method that attributes image–text similarity by grouping patches into coherent clusters, masking them, and evaluating relative changes in model predictions. CCI sets a new state of the art on faithfulness benchmarks, surpassing prior methods by large margins; for example, it yields more than a twofold improvement on the deletion-AUC metric for MS COCO retrieval. We further propose that CCI when combined with GroundedSAM, automatically categorizes predictions as foreground or background-driven, providing a crucial diagnostic ability. Existing benchmarks such as CounterAnimals, however, rely solely on accuracy and implicitly attribute all performance degradation to background correlations. Our analysis shows this assumption to be incomplete, since many errors arise from viewpoint variation, scale shifts, and fine-grained object confusions. To disentangle these effects, we introduce COVAR, a benchmark that systematically varies object foregrounds and backgrounds. Leveraging CCI with COVAR, we conduct a comprehensive evaluation of eighteen CLIP variants, providing both methodological advances and empirical evidence that chart a path toward more robust vision–language models.

## 1 INTRODUCTION

Contrastive vision–language models (VLMs) such as CLIP (Radford et al., 2021) demonstrate strong generalization in tasks including zero-shot classification, retrieval (Luo et al., 2021), and open-vocabulary recognition (Li et al., 2022; 2021) across diverse domains (Kim et al., 2022; Liu et al., 2024). Despite this success, they remain vulnerable to spurious correlations (Wang et al., 2024; Yang et al., 2023; Xu et al., 2025), that is, associations driven by dataset biases rather than true semantic grounding (Geirhos et al., 2020). For instance, in Figure 1(a), CLIP predicts water ouzel by relying on the surrounding water instead of the object. Such correlations reduce robustness under distribution shifts (Chen et al., 2025a; Koddenbrock et al., 2025; Varma et al., 2024).

Recent work shows spurious correlations in VLMs often stem from dataset-specific artifacts, with object–background priors constituting a dominant source (e.g., zebras in grasslands, sharks in water) (Tian et al., 2025; Lu et al., 2025; Yang et al., 2025; Wang et al., 2024; Varma et al., 2024). A notable attempt to quantify the role of background context is the CounterAnimals (CA) benchmark (Wang et al., 2024), which partitions images into "easy" and "hard" sets based on CLIP's accuracy, with the latter intended to probe sensitivity to atypical backgrounds. However, this operationalization is limited: declines in accuracy need not unambiguously signify reliance on background features, while elevated accuracy cannot be presumed to reflect object-centric reasoning.

Using our interpretability method CCI (Section 3), we analyze CLIP's (ViT-B/16) behavior on the CA dataset and show that accuracy-based partitioning is insufficient for diagnosing background sensitivity. As illustrated in Figure 1(a), correct predictions may rely on background cues (e.g., *water ouzel* identified via surrounding *water*), while errors can occur despite object-focused attention (e.g., *jaguar* misclassified as *cheetah*). Quantitative results reinforce this: although 32.1% of the easy set and 46.6% of the hard set are misclassified, the proportion attributable to background reliance remains nearly identical (6.23% vs. 7.26%, *BG-Er*, Figure 1(d)). Instead, most failures

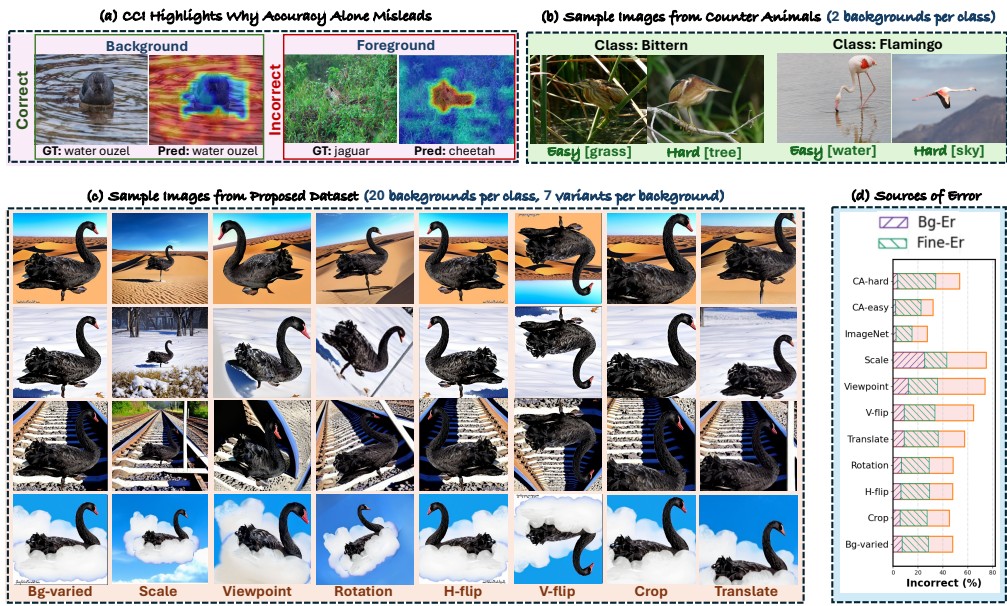

Figure 1: (a) Image and CCI maps for CLIP's prediction (red–blue heatmap, red = stronger attention). (b) Samples from easy and hard sets of CA. (c) Example of dataset curation in COVAR. (d) Proportion of different sources of errors in CA, ImageNet and COVAR subsets.

arise from fine-grained confusions (*Fine-Er*), indicating that raw accuracy is an unreliable proxy for disentangling object versus background-driven errors.

These findings highlight structural limitations of CA. By collapsing diverse visual variation into a binary easy–hard split, it overlooks heterogeneity in viewpoint, scale, pose, and composition. Errors therefore often reflect non-background factors; for example, failures on the *flamingo* class (Figure 1(b)) may stem from contrasting poses across easy–hard splits rather than background context. As shown in Figure 1(d), background-driven errors (*BG-Er*) form only a small fraction overall. Consequently, CA's coarse partitioning and absence of fine-grained annotations restrict its diagnostic capacity and preclude precise assessment of CLIP's dependence on background correlations.

In summary, current benchmarks for spurious background correlations are deficient in two respects: (i) accuracy is an inadequate proxy, e.g., conflating background reliance with fine-grained class confusions, and (ii) effective evaluation requires controlled, diverse variation in visual factors to both isolate individual effects and expose a broader spectrum of failure modes.

To address these gaps, our first contribution is a training-free method, called Cluster-based Concept Importance (CCI), to interpret CLIP's predictions. CCI identifies the image regions driving image–text similarity by grouping patches into semantically coherent clusters, systematically masking them, and measuring changes in model predictions relative to the unmasked input (e.g. Figure 1, first row). In contrast to gradient-based approaches Zhao et al. (2025), which frequently yield noisy and fragmented pixel-level saliency maps, CCI identifies coherent and semantically meaningful regions, thereby enabling direct region-level interpretability (Figure 3). This property is crucial for our analysis, as CCI disentangles background and object contributions, providing a principled basis for diagnosing shortcut reliance and robustness failures in CLIP.

Our second contribution is COVAR (**CO**ntrolled **VAR**iants), a dataset developed for systematically evaluating models' reliance on spurious correlations. In contrast to the CA benchmark, which provides only two background variants and ignores other visual dimensions, COVAR introduces fine-grained and explicitly controlled variations. For each class, we begin with a base image (e.g., a *swan*, see Figure 1) and generate paired variants that independently perturb background, viewpoint, scale, translation, flip, rotation, and crop. Through this design, multiple factors shaping model behavior are explicitly represented and disentangled, thereby enabling rigorous, factor-wise evaluation.

As our third contribution, we benchmark 18 CLIP variants on CCI and our proposed dataset COVAR (Section 4) to assess their reliance on spurious correlations. We present results across eight subsets of COVAR, identify the dominant influences on model behavior, and provide insights for mitigating these vulnerabilities.

## 2 RELATED WORKS

Research on vision–language models spans *interpretability*, which explains predictive mechanisms, and *robustness evaluation*, which probes behavior under distribution shifts and challenging conditions. These are intertwined: interpretability reveals the cues models rely on, while robustness tests whether such dependencies yield failures like spurious correlations.

**Interpretability in Vision-Language Models:** Early interpretability efforts applied methods such as attention rollout (Abnar & Zuidema, 2020), which aggregates attention weights across layers, and gradient-based techniques like GradCAM (Selvaraju et al., 2017) and GAME (Chefer et al., 2021a), which compute feature importance through gradients. With the advent of CLIP (Radford et al., 2021) and successors like BLIP (Li et al., 2022) and ALBEF (Li et al., 2021), researchers developed VLM-specific techniques: Grad-ECLIP (Zhao et al., 2025) extends gradient-based attribution to multimodal settings, CLIPSurgery (Li et al., 2023) identifies important regions through similarity masking, MaskCLIP (Zhou et al., 2022) performs token-level masking to localize influential regions, ECLIP (Chefer et al., 2021b) leverages gradient-weighted attention, and M2iB (Wang et al., 2023) employs perturbation-based explanations. Perturbation-driven methods such as RISE (Petsiuk et al., 2018) further provide model-agnostic saliency through random masking. More recent concept-based approaches like SpLiCE (Bhalla et al., 2024) learn sparse concept embeddings, while others (Kim et al., 2024) use foundation models for concept discovery, building on earlier methods like Concept Bottleneck Models (Koh et al., 2020). Existing methods yield noisy, low-level attributions lacking semantic coherence. CCI overcomes these limitation with concept-level attributions.

**Benchmarks for Robustness and Spurious Correlations:** Complementing interpretability work, robustness benchmarks have evolved from broad evaluations of adversarial robustness (Goodfellow et al., 2014) and distribution shifts (Hendrycks & Dietterich, 2019) to targeted assessments of specific weaknesses. ImageNet-A/R/C (Hendrycks et al., 2021) and WILDS (Koh et al., 2021) highlighted vulnerabilities to natural adversarial examples and domain shifts, while synthetic datasets like ObjectNet (Barbu et al., 2019) and 3D-Common (Kar et al., 2022) enabled systematic testing of pose and context. For VLMs, specialized benchmarks have emerged: Winoground (Thrush et al., 2022) probes compositional reasoning, and Waterbirds (Sagawa et al., 2019) exposes background bias in bird classification. Most recently, CA (Wang et al., 2024) investigated CLIP's background sensitivity but, by relying on accuracy drops as a proxy. We extend this line of work by isolating visual factors and applying interpretability to distinguish spurious correlations from other errors.

## 3 CONCEPT CLUSTER IMPORTANCE

We present **Concept Cluster Importance (CCI)**, a training-free interpretability method for CLIP models that quantifies the contribution of semantically coherent visual concepts to image–text similarity scores. CCI operates entirely at inference time, requiring no model modification or retraining.

**Patch Embedding Clustering.** We focus on the patch embeddings $\mathbf{X} = \{\mathbf{z}_i\}_{i=1}^N$, which encode localized semantics. To extract coherent visual concepts, we perform K-means clustering over $\mathbf{X}$, yielding $\mathcal{C} = \{C_1, \ldots, C_K\}$, where each cluster aggregates semantically similar patches (See Figure 2 for examples with $K = 7$, where distinct colors denote different clusters).

**Preliminaries.** Given an input image $I$, the CLIP image encoder (e.g., a Vision Transformer) processes $I$ into a sequence of token embeddings $\mathbf{Z} = [\mathbf{z}_{\text{CLS}}, \mathbf{z}_1, \mathbf{z}_2, \ldots, \mathbf{z}_N] \in \mathbb{R}^{(N+1) \times d}$ where $\mathbf{z}_{\text{CLS}} \in \mathbb{R}^d$ is the global [CLS] embedding used for final image representation, and $\{\mathbf{z}_i\}_{i=1}^N$ are patch embeddings corresponding to $N$ fixed-size patches extracted from the image.

**Attention Attenuation with Cluster Masks.** The CLIP image encoder contains $L$ transformer layers, each with self-attention matrices $A^{(l)} \in \mathbb{R}^{(N+1) \times (N+1)}$, where the first token corresponds to the CLS embedding and the rest correspond to patches.

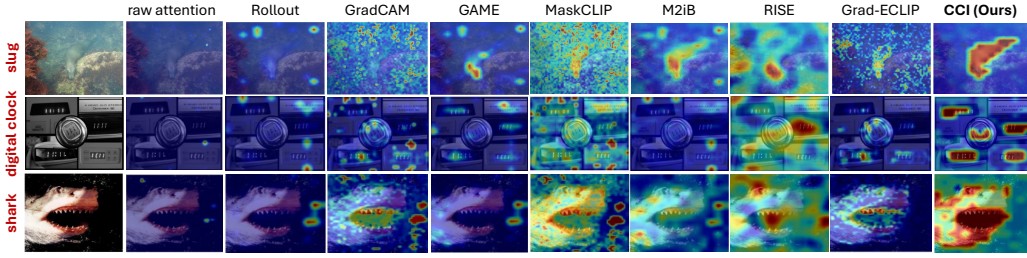

Figure 3: Qualitative comparison of CCI against baseline interpretability methods.

To measure the contribution of cluster $C_k$, we construct a binary mask $m_k(j)$ that indicates whether patch $j$ belongs to cluster $C_k$:

$$m_k(j) = \begin{cases} 1, & j \in C_k, \\ 0, & \text{otherwise.} \end{cases}$$

The attention logits are then modified before softmax:

$$\hat{A}_k^{(l)}(i,j) = \begin{cases} A^{(l)}(i,j), & m_k(j) = 0, \\ -\infty, & m_k(j) = 1. \end{cases}$$

This masking is applied at every transformer layer and head, preventing the CLS token from aggregating information from the patches corresponding to the selected cluster.

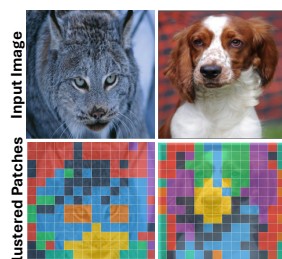

Figure 2: Patch clusters.

**Importance Scoring via Similarity Drop.** Let $\mathbf{z}_{\text{CLS}}$ be the original final CLS embedding, and $\mathbf{t}$ the corresponding text embedding. The original similarity score is $s = \cos(\mathbf{z}_{\text{CLS}}, \mathbf{t}) = \frac{\mathbf{z}_{\text{CLS}}^{\top}\mathbf{t}}{\|\mathbf{z}_{\text{CLS}}\|\|\mathbf{t}\|}$. After attention attenuation for cluster $k$, we obtain a modified CLS embedding $\hat{\mathbf{z}}_{\text{CLS},k}$ and similarity $s_k = \cos(\hat{\mathbf{z}}_{\text{CLS},k}, \mathbf{t})$.

The relative importance of cluster $C_k$ is quantified by the similarity drop $\Delta s_k = s - s_k$. We normalize the importance scores as $w_k = \frac{\Delta s_k}{\sum_{j=1}^{K} \Delta s_j}$, and compute the spatial importance map $S \in \mathbb{R}^{W \times H}$ as a weighted sum of cluster masks $S = \sum_{k=1}^{K} w_k \cdot m_k$. The map $S$ identifies regions contributing most to the similarity score and is visualized as a heatmap overlay on $I$.

### 3.1 CCI Results

We evaluate CCI against a diverse set of baselines spanning gradient-based, attention-based, and perturbation-based techniques: Attention Rollout (Abnar & Zuidema, 2020), GradCAM (Selvaraju et al., 2017), GAME (Chefer et al., 2021a), MaskCLIP (Zhou et al., 2022), M2iB (Wang et al., 2023), RISE (Petsiuk et al., 2018), and Grad-ECLIP (Zhao et al., 2025). Unless otherwise specified, all experiments use CLIP with a `ViT-B/16` image encoder, and maps are computed with respect to the ground-truth class label.

**Qualitative Comparison with Baselines.** Figure 3 compares attention maps from CCI and baseline methods. CCI consistently produces coherent, object-aligned heatmaps, whereas baselines yield sparse or noisy patterns. For instance, in the *slug* image (row 1), CCI captures the entire object while baselines highlight scattered regions; in the *digital clock* (row 2), CCI sharply localizes the digits on the clock face, relevant to CLIP's prediction, unlike baselines that miss or misfocus; and in the *shark* example (row 3), CCI emphasizes the teeth-features, whereas baselines diffuse attention across irrelevant regions. Additional examples are provided in supplementary (Appendix A.1). We also show results with variants other than CLIP `ViT-B/16` in supplementary (Appendix A.2). These results underscore CCI's key strengths: the clustering of semantically related patches into coherent concepts and the principled use of similarity-drop scoring to quantify their predictive contribution, together enabling precise, concept-level visualizations.

**Quantitative Comparison.** Consistent with prior work, we evaluate the faithfulness of CCI using deletion and insertion metrics (Samek et al., 2016). CCI produces a patch-level importance

Table 1: Faithfulness evaluation of image explanations on *ImageNet* validation: AUC of Deletion/Insertion curves using Top-1 (@1) or Top-5 (@5) accuracy, with either ground-truth or predicted labels as CLIP text input.

| Method | Deletion ↓ | | | | Insertion ↑ | | | |
|---|---|---|---|---|---|---|---|---|
| | Ground-truth | | Prediction | | Ground-truth | | Prediction | |
| | @1 | @5 | @1 | @5 | @1 | @5 | @1 | @5 |
| raw attention | 0.3831 | 0.6239 | - | - | 0.2492 | 0.4195 | - | - |
| Rollout | 0.4082 | 0.6556 | - | - | 0.2803 | 0.4665 | - | - |
| Grad-CAM | 0.3417 | 0.5628 | 0.3518 | 0.5817 | 0.2682 | 0.4454 | 0.2526 | 0.4206 |
| GAME | 0.3356 | 0.5734 | 0.3497 | 0.5938 | 0.3611 | 0.5636 | 0.3425 | 0.5384 |
| MaskCLIP | 0.2848 | 0.4885 | 0.2886 | 0.4957 | 0.3335 | 0.5351 | 0.3275 | 0.5267 |
| CLIPSurgery | 0.3115 | 0.5235 | 0.3217 | 0.5412 | 0.3832 | 0.6021 | 0.3727 | 0.5719 |
| M2IB | 0.3630 | 0.5953 | 0.3633 | 0.5951 | 0.3351 | 0.5411 | 0.3347 | 0.5410 |
| Grad-ECLIP w/o $\lambda_i$ | 0.2535 | 0.4379 | 0.2634 | 0.4568 | 0.3715 | 0.5831 | 0.3528 | 0.5556 |
| Grad-ECLIP | 0.2464 | 0.4272 | 0.2543 | 0.4420 | 0.3838 | 0.5993 | 0.3672 | 0.5749 |
| **CCI (Ours)** | **0.1809** | **0.3276** | **0.1789** | **0.3318** | **0.4175** | **0.6518** | **0.3893** | **0.6201** |

Table 2: Evaluation of **image** explanation faithfulness on *MS COCO* image-text retrieval (*Karpathy's split*) val-set: AUC for Deletion and Insertion curves for performance on image retrieval (IR) and text retrieval (TR) tasks.

| Method | Deletion↓ | | | | Insertion↑ | | | |
|---|---|---|---|---|---|---|---|---|
| | IR | | TR | | IR | | TR | |
| | @1 | @5 | @1 | @5 | @1 | @5 | @1 | @5 |
| raw attention | 0.1708 | 0.3554 | 0.1923 | 0.3720 | 0.1247 | 0.2552 | 0.1544 | 0.2969 |
| Rollout | 0.1948 | 0.3946 | 0.2268 | 0.4238 | 0.1294 | 0.2932 | 0.1753 | 0.3503 |
| Grad-CAM | 0.1717 | 0.3502 | 0.2161 | 0.4008 | 0.1027 | 0.2216 | 0.1152 | 0.2327 |
| GAME | 0.1706 | 0.3552 | 0.1982 | 0.3800 | 0.1537 | 0.3083 | 0.2097 | 0.3735 |
| MaskCLIP | 0.1321 | 0.2841 | 0.1516 | 0.2949 | 0.1423 | 0.2953 | 0.1891 | 0.3514 |
| CLIPSurgery | 0.1794 | 0.3652 | 0.2381 | 0.4292 | 0.1419 | 0.2941 | 0.1771 | 0.3384 |
| M2IB | 0.1797 | 0.3671 | 0.2057 | 0.3905 | 0.1469 | 0.3004 | 0.2058 | 0.3691 |
| Grad-ECLIP w/o $\lambda_i$ | 0.1390 | 0.2940 | 0.1827 | 0.3386 | 0.1403 | 0.2895 | 0.1735 | 0.3279 |
| Grad-ECLIP | 0.1246 | 0.2670 | 0.1550 | 0.2933 | 0.1576 | 0.3203 | 0.2056 | 0.3761 |
| **CCI (Ours)** | **0.0650** | **0.1056** | **0.0677** | **0.1184** | **0.1812** | **0.3513** | **0.2224** | **0.3943** |

map (14×14 for `ViT-B/16`), which is upsampled to 224×224 to assign pixel-level scores. Pixels are ranked by importance; in *deletion*, top-ranked pixels are iteratively replaced with random noise, while in *insertion*, they are progressively revealed from a blank canvas. At each step, ∼0.5% of pixels are modified, over 100 steps, cumulatively altering about half the image. The model's top-1 and top-5 accuracy is tracked at every step, and the area under the resulting curves (AUC) is used as a summary measure: lower AUC for deletion and higher AUC for insertion indicate causal influence of the highlighted regions. We report these scores on ImageNet-1K classification (Deng et al., 2009)) and MS COCO cross-modal retrieval (Lin et al., 2014).

Across both datasets, CCI consistently outperforms all baselines. On ImageNet, deletion curves in Figure 4 drop sharply when removing regions identified by CCI, while insertion curves recover accuracy substantially faster, confirming that the highlighted regions capture the core evidence leveraged by CLIP. These trends are reflected in the AUC scores (Table 1), where CCI achieves state-of-the-art performance across Top-1 and Top-5 metrics. The same

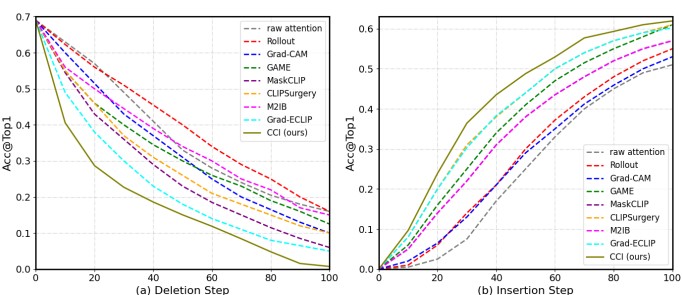

Figure 4: Deletion and insertion curves demonstrating CCI's quantitative superiority in identifying decision-relevant regions.

holds for image–text retrieval (Table 2), where CCI delivers state-of-the-art results for both image and text retrieval on COCO. Notably, in deletion, CCI attains over two-fold error reduction (0.2670 → 0.1056) in Top-5 IR, over the second-best Grad-ECLIP method. Collectively, these results show that CCI yields faithful, generalizable attributions of CLIP's decisions.

**Understanding CLIP's Failure Modes:** We use CCI to analyze CLIP's zero-shot predictions by visualizing attention maps with respect to the predicted class (Figure 5, ImageNet-1k in part a and CA in part b), focusing on misclassifications to expose error sources. Several failures are *foreground-driven*: in part a, row 1, a chimpanzee is misclassified as a siamang despite correct focus on the face; in row 2, attention is misdirected to a bucket, leading to error; and in row 3, CLIP fixates on a squirrel rather than the target class. In CA (part b, rows 2–3), CCI likewise reveals attention on the foreground, but errors arise from clutter, occlusion, or subtle visual distinctions.

Other errors are *background-driven*: in part a, row 4, attention to the grassy field results in *croquet ball*, while in part b, row 1, focus on background produces *water snake*. Even under occlusion or partial visibility (part b, rows 2 and 4), CCI consistently highlights the object, whereas baselines scatter attention across irrelevant regions. Overall, CCI accurately predicts CLIP's attention, offering interpretable insights, with clarity unmatched by existing methods.

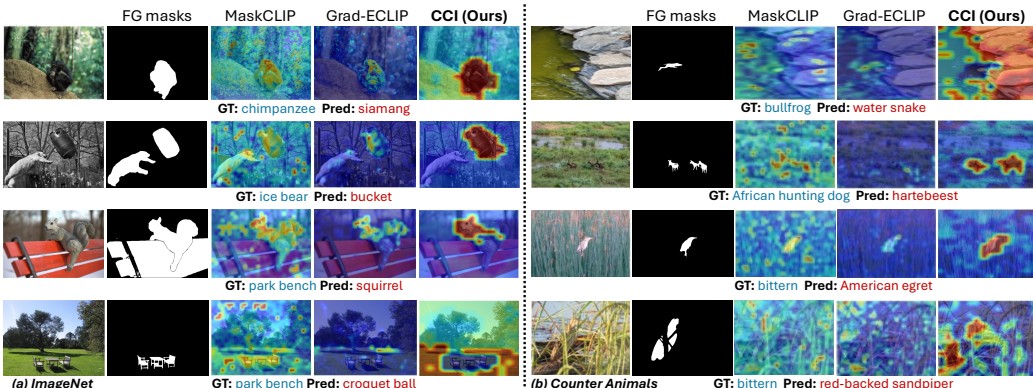

Figure 5: CCI analysis of CLIP failures on (a) ImageNet and (b) Counter Animals. Rows show the image, ground-truth foreground (FG) mask obtained using GroundedSAM, and attribution maps from MaskCLIP, Grad-ECLIP, and CCI, with ground-truth (GT) and predicted (Pred) labels below.

**Diagnosing Error Sources with CCI:** While prior sections qualitatively showed that CLIP's errors extend beyond background bias to factors such as viewpoint, occlusion, and fine-grained confusion, here we use CCI to systematically measure the role of background cues. Using GroundedSAM (Ren et al., 2024), we obtain ground-truth foreground (FG) and background (BG) masks for ImageNet-1k and CA datasets (details are given in supplementary, Appendix A.3). CCI heatmaps are then computed with respect to CLIP's predicted class, and IoU overlap with FG/BG masks is used to classify predictions as *foreground-driven* (*FG-Er*) or *background-driven* (*BG-Er*).

For *foreground-driven* errors, we further identify cases of *fine-grained confusion* (*Fine-Er*). Using GPT-4o (prompt in Appendix A.4.1), we assess whether the predicted and ground-truth categories are visually similar (e.g., *siamang* vs. *chimpanzee*) or unrelated, thereby distinguishing subtle fine-grained inter-class confusions from more severe errors caused by distractors or other factors.

Figure 1(d) summarizes errors: the first three rows show CA-hard, CA-easy, and ImageNet respectively. *BG-Er* constitute only a small fraction (9.1% on ImageNet, 6.7% on CA), with similar rates across CA's *easy* and *hard* sets, questioning the assumption that accuracy gaps primarily reflect background correlations. In contrast, a substantial portion of errors arise from *Fine-Er* (46.6% on ImageNet-1k, 60.4% on CA), underscoring the need to look beyond the background cues.

## 4 COVAR: A NEW BENCHMARK

As discussed in Section 1 and shown in our CCI analyses, the CA benchmark is limited, offering only coarse background variation and no control over viewpoint, scale, flip, or crop. To address these gaps, we introduce COVAR, where each object is placed into multiple backgrounds and, for every such instance, its appearance is systematically varied along several visual factors.

### 4.1 CREATING BACKGROUND VARIATIONS

The objective is to synthesize diverse backgrounds for a given object. Unlike CA, which has limited diversity (45 animal classes), COVAR spans semantically and visually varied categories, from non-living objects (e.g., airships, locomotives) to living entities (e.g., birds, reptiles). A complete class list with ID mappings is provided in the Appendix A.4.2.

We select 33 classes from ImageNet and sample 50 images per class. Using the Emu2 image editing model Sun et al. (2024), each image is synthesized across 20 curated background types (via GPT-4o (Hurst et al., 2024) prompts; see Appendix A.4.3) spanning outdoor (e.g., beach, railway track) and indoor (e.g., living room, kitchen) contexts, resulting in 1000 images per class. Figure 1(c) illustrates this process with a swan across four different backgrounds.

We evaluate the 33,000 background-varied (Bg-varied) images using CLIP `ViT-B/16` in zero-shot mode over the full 1000 class-labels from ImageNet. Figure 6 reports the average per-class accuracy drop relative to the original ImageNet images. While the mean drop is 23.78, the effect is highly uneven: classes 11, 22, and 31 show little-to-no decline, whereas classes

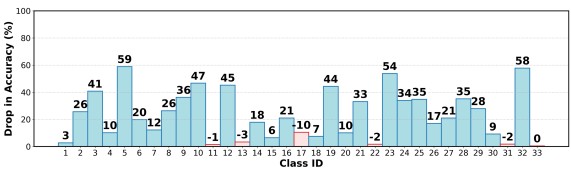

Figure 6: Class-wise accuracy drops for Bg-varied set.

5, 23, and 32 suffer drops exceeding 50%, indicating that some categories are robust to background variation while others are strongly background-dependent. We show similar per-class accuracy drops for CLIP variants other than `ViT-B/16` in the supplementary material (Appendix A.5).

## 4.2 EXTENDING WITH STRUCTURED VARIANTS

To study additional factors influencing CLIP's accuracy, we extend each of the 33,000 Bg-varied images into 11 structured transformations: four different scales, two different viewpoints, horizontal and vertical flips, and single versions of translation, crop, and rotation (see Appendix A.6 for implementation details). Figure 1(c) illustrates this setup: the first column shows the object in varied backgrounds, while the remaining columns show transformed images. Together with the Bg-varied images, this expansion results in 396,000 samples, constituting the complete COVAR dataset.

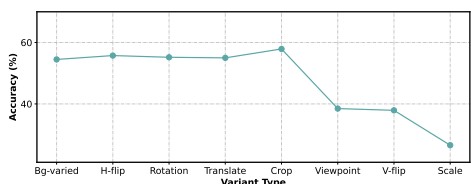

Figure 7: Subset-wise accuracy drops

We conduct zero-shot classification with CLIP `ViT-B/16` on eight COVAR subsets (bg-varied and seven transformations). Figure 7 shows that when compared with Bg-varied set, factors such as scale, v-flip and viewpoint cause steep declines. Hflip, rotation and translate lead to little or no degradation, crop leads to minor gains.

We evaluate class-wise accuracy drops (Figure 8), for each class, accuracies are computed across the eight subsets, averaged, and compared to the accuracy on the original ImageNet images. We also record the drop for the subset with maximum decline. The observed drops are substantially larger than those for background alone. Notably, even when average drops are modest, individual subsets can show severe degradation. For example, for class 22 the average drop is 10% while one subset exceeds 50%. Comprehensive subset-wise results for all classes are provided in the supplementary material (Appendix A.7).

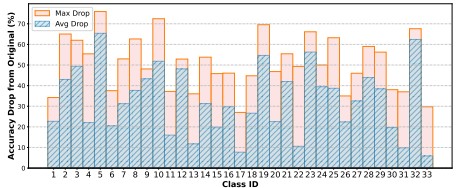

Figure 8: Class-wise average and maximum accuracy drops across variants.

**Diagnosing error sources.** We systematically analyze *BG-Er* and *Fine-Er* across all COVAR subsets, benchmarking against CA and original ImageNet (Figure 1(d)). The Bg-varied subset exhibits a notable rise in *BG-Er* relative to CA (6.7 to 15.6%). Scale and viewpoint variations further elevate both overall error rates and the proportion of *BG-Er* failures, with scale showing the strongest effect, indicating increased reliance on background cues under these transformations. In contrast, vertical flip, horizontal flip, translation, and crop do not exhibit such increases. Across all subsets, however, *Fine-Er* remains the dominant source of error.

## 5 EVALUATION AND DISCUSSION

We benchmark CLIP variants from OpenCLIP (Ilharco et al., 2021) and OpenAI's original models (Radford et al., 2021) on COVAR, varying backbone size, patch resolution, and pretraining data (e.g., DataComp (Gadre et al., 2023), LAION (Schuhmann et al., 2022), DFN (Fang et al., 2023), WebLI (Chen et al., 2022)). We also evaluate SigLIP variants (Zhai et al., 2023; Tschannen et al., 2025). Tables 3 and 4 present classification accuracy along with *BG-Er* and *Fine-Er*.

Table 3: CLIP variant accuracies across subsets of COVAR; final column shows overall average.

| Name | Method | | | Bg-varied | H-flip | Translate | Crop | V-flip | Rotation | Viewpoint | Scale | Avg |
|---|---|---|---|---|---|---|---|---|---|---|---|---|
| | Patch | Res | Dataset | Acc | Acc | Acc | Acc | Acc | Acc | Acc | Acc | Acc |
| ViT-B | 32 | 256 | DataComp-1B | 55.8 | 55.2 | 55.8 | 59.4 | 32.2 | 44.6 | 28.5 | 24.2 | 44.5 |
| ViT-B | 16 | 224 | DataComp-1B | 55.7 | 54.3 | 53.7 | 56.5 | 37.5 | 44.9 | 27.7 | 27.2 | 44.7 |
| ViT-L | 14 | 224 | DataComp-1B | 62.2 | 61.3 | 60.7 | 62.3 | 48.3 | 54.9 | 30.3 | 32.6 | 51.6 |
| ViT-L | 14 | 224 | LAION-2B | 59.7 | 58.9 | 60.0 | 60.2 | 43.2 | 53.9 | 32.2 | 30.1 | 49.8 |
| ViT-H | 14 | 224 | LAION-2B | 60.2 | 59.2 | 59.4 | 60.9 | 45.3 | 54.2 | 30.1 | 31.0 | 50.0 |
| ViT-bigG | 14 | 224 | LAION-2B | 61.6 | 62.1 | 62.2 | 63.8 | 45.7 | 56.5 | 34.2 | 33.5 | 52.5 |
| ViT-L | 14 | 224 | DFN-2B | 59.1 | 57.8 | 58.0 | 59.5 | 41.8 | 51.3 | 28.0 | 31.1 | 48.3 |
| ViT-SO-SigLIP2 | 14 | 224 | WebLI | 63.3 | 62.7 | 62.4 | 63.6 | 53.0 | 58.6 | 34.1 | 36.9 | 54.3 |
| ViT-B-SigLIP | 16 | 256 | WebLI | 57.5 | 56.8 | 56.6 | 57.4 | 38.7 | 48.9 | 27.9 | 29.1 | 46.6 |
| ViT-B-SigLIP2 | 16 | 384 | WebLI | 64.1 | 63.3 | 63.4 | 63.8 | 48.1 | 59.9 | 31.0 | 37.8 | 53.9 |
| ViT-B-SigLIP2 | 32 | 256 | WebLI | 54.8 | 53.9 | 55.0 | 58.2 | 30.9 | 41.6 | 26.0 | 25.6 | 43.2 |
| ViT-B-SigLIP2 | 16 | 512 | WebLI | 64.9 | 63.8 | 63.9 | 64.1 | 49.3 | 60.6 | 30.5 | 36.4 | 54.2 |
| ViT-H-qgelu | 14 | 378 | DFN-5B | 65.2 | 65.4 | 65.5 | 65.7 | 55.0 | 61.7 | 36.7 | 39.3 | 56.8 |
| ViT-H-qgelu | 14 | 224 | DFN-5B | 63.2 | 62.9 | 63.3 | 64.1 | 50.9 | 57.4 | 35.9 | 38.2 | 54.5 |
| ViT-B | 16 | 224 | OpenAI | 52.2 | 52.3 | 51.7 | 54.8 | 35.2 | 42.6 | 26.1 | 25.0 | 42.5 |
| ViT-B | 32 | 224 | OpenAI | 47.9 | 48.0 | 46.0 | 48.3 | 26.8 | 30.1 | 21.8 | 22.1 | 36.4 |
| ViT-L | 14 | 224 | OpenAI | 56.5 | 56.7 | 56.0 | 57.4 | 46.9 | 49.2 | 26.7 | 28.3 | 47.2 |
| ViT-L | 14 | 336 | OpenAI | 57.6 | 57.8 | 57.2 | 58.4 | 49.5 | 53.1 | 27.3 | 31.8 | 49.1 |

Table 4: *BG-Er* and *Fine-Er* across different subsets of COVAR.

| Name | Method | | | Bg-varied | | H-flip | | Translate | | Crop | | V-flip | | Rotation | | Viewpoint | | Scale | |
|---|---|---|---|---|---|---|---|---|---|---|---|---|---|---|---|---|---|---|---|
| | Patch | Res | Dataset | BG-Er | Fine-Er | BG-Er | Fine-Er | BG-Er | Fine-Er | BG-Er | Fine-Er | BG-Er | Fine-Er | BG-Er | Fine-Er | BG-Er | Fine-Er | BG-Er | Fine-Er |
| ViT-B | 32 | 256 | DComp-1B | 23.6 | 40.9 | 21.2 | 44.6 | 18.9 | 46.2 | 18.5 | 47.7 | 22.6 | 33.3 | 21.3 | 39.5 | 22.9 | 29.5 | 50.7 | 16.1 |
| ViT-B | 16 | 224 | DComp-1B | 14.2 | 49.3 | 11.7 | 52.7 | 11.3 | 53.4 | 11.9 | 52.2 | 13.8 | 42.6 | 12.4 | 50.4 | 15.7 | 33.5 | 30.5 | 28.4 |
| ViT-L | 14 | 224 | DComp-1B | 15.7 | 51.8 | 13.5 | 56.0 | 12.4 | 56.6 | 12.2 | 56.6 | 15.2 | 50.2 | 14.5 | 53.9 | 17.1 | 36.0 | 30.2 | 31.2 |
| ViT-L | 14 | 224 | LAION-2B | 16.9 | 49.8 | 14.0 | 55.0 | 13.8 | 53.2 | 12.4 | 55.4 | 15.4 | 44.3 | 16.2 | 51.4 | 17.1 | 33.5 | 35.5 | 27.8 |
| ViT-H | 14 | 224 | LAION-2B | 16.4 | 54.8 | 13.6 | 59.4 | 13.0 | 58.7 | 15.1 | 56.3 | 15.9 | 49.6 | 14.8 | 55.2 | 18.2 | 35.1 | 33.8 | 30.9 |
| ViT-bigG | 14 | 224 | LAION-2B | 17.8 | 51.2 | 16.0 | 54.3 | 16.4 | 53.6 | 17.2 | 52.3 | 15.4 | 48.3 | 17.4 | 54.0 | 19.6 | 33.0 | 32.7 | 29.7 |
| ViT-L | 14 | 224 | DFN-2B | 15.7 | 50.2 | 13.7 | 53.2 | 14.2 | 54.1 | 15.5 | 52.3 | 15.2 | 45.6 | 13.5 | 54.4 | 17.4 | 34.7 | 29.6 | 31.8 |
| ViT-SO-SigLIP2 | 14 | 224 | WebLI | 25.3 | 47.8 | 22.7 | 52.8 | 21.2 | 53.1 | 20.0 | 53.8 | 23.0 | 51.7 | 25.4 | 51.4 | 27.0 | 32.7 | 42.3 | 29.3 |
| ViT-B-SigLIP | 16 | 256 | WebLI | 16.5 | 49.2 | 13.5 | 53.6 | 13.0 | 53.8 | 13.0 | 54.4 | 15.2 | 44.3 | 14.4 | 52.6 | 17.1 | 33.8 | 33.6 | 26.7 |
| ViT-B-SigLIP2 | 16 | 384 | WebLI | 19.8 | 51.5 | 17.5 | 55.2 | 15.9 | 56.7 | 16.6 | 55.2 | 19.7 | 49.9 | 20.2 | 54.5 | 25.4 | 33.1 | 36.8 | 28.3 |
| ViT-B-SigLIP2 | 32 | 256 | WebLI | 36.0 | 33.6 | 35.8 | 35.0 | 37.2 | 34.6 | 31.2 | 42.7 | 36.1 | 24.5 | 32.3 | 24.7 | 34.7 | 20.8 | 45.7 | 7.7 |
| ViT-B-SigLIP2 | 16 | 512 | WebLI | 20.3 | 51.8 | 18.1 | 55.2 | 16.7 | 55.8 | 17.2 | 53.1 | 20.3 | 50.1 | 20.2 | 53.8 | 24.8 | 31.9 | 39.3 | 22.3 |
| ViT-H-qgelu | 14 | 378 | DFN-5B | 15.9 | 54.4 | 13.4 | 57.8 | 13.4 | 57.8 | 12.7 | 58.5 | 14.9 | 54.4 | 15.4 | 57.1 | 16.8 | 36.3 | 30.4 | 33.1 |
| ViT-H-qgelu | 14 | 224 | DFN-5B | 16.2 | 51.3 | 13.8 | 55.7 | 13.6 | 54.8 | 15.4 | 54.2 | 15.6 | 49.2 | 15.0 | 52.5 | 17.4 | 34.7 | 29.4 | 23.2 |
| ViT-B | 16 | 224 | OpenAI | 15.6 | 43.7 | 13.8 | 48.2 | 14.0 | 48.8 | 13.1 | 48.0 | 14.2 | 37.9 | 16.0 | 45.1 | 16.8 | 31.3 | 33.9 | 23.7 |
| ViT-B | 32 | 224 | OpenAI | 14.0 | 37.3 | 12.0 | 41.5 | 13.7 | 42.0 | 12.6 | 46.3 | 13.2 | 30.2 | 15.3 | 34.4 | 14.5 | 26.4 | 38.2 | 21.4 |
| ViT-L | 14 | 224 | OpenAI | 17.1 | 44.3 | 14.9 | 48.2 | 15.2 | 48.1 | 16.8 | 45.2 | 15.6 | 44.6 | 16.8 | 46.0 | 16.7 | 34.3 | 30.1 | 12.1 |
| ViT-L | 14 | 336 | OpenAI | 15.3 | 46.4 | 12.8 | 50.8 | 13.3 | 49.7 | 11.7 | 50.9 | 13.4 | 48.1 | 15.3 | 49.2 | 14.4 | 35.4 | 28.4 | 30.4 |

**Overall performance:** Among all models, `ViT-H-qgelu(DFN-5B)` at `378px` achieves the highest average accuracy of 56.8% (Table 3). `ViT-B-SigLIP2` performs strongly at `384px` and `512px`, while `ViT-SO-SigLIP2` maintains competitive accuracy even at `224px`, underscoring its training efficiency. Larger models such as `ViT-bigG` also show reasonable performance.

**Performance across eight subsets:** On the Bg-varied set, most models retain accuracies above 55%. Cropping does not hinder performance and often improves it, while H-flip and Translation cause only minor declines (e.g., large models such as `ViT-bigG` remains above 62%). Rotation produces minor drops, V-flip somewhat larger, and Viewpoint and Scale lead to the most severe declines. Across all perturbations, larger models consistently surpass smaller ones; for example, under Rotation, `ViT-H-qgelu(378px)` achieves 57.4% accuracy versus 30.1% for `ViT-B/32(DataComp-1B)`, underscoring the benefits of greater capacity and training scale.

Among all variants, Scale is the most challenging. `ViT-L/14(DataComp-1B)` drops from 62.2% on Bg-varied to 32.6% under Scale, while even the stronger `ViT-H-qgelu(378px)` declines from 65.2% to 39.3%. Viewpoint changes also yield substantial drops (e.g., `ViT-L/14` to 30.3%), though unlike Scale they do not coincide with major increases in *BG-Er* (Table 4). The first row in Appendix A.8 demonstrates how reduction in object scale may cause reliance on background.

**Background reliance.** Accuracy drops alone do not explain why models fail. Reducing Scale not only lowers accuracy but also nearly doubles *BG-Er* relative to the Bg-varied subset (Table 4), reaching 50.7% for `ViT-B/32(DataComp-1B)`. By contrast, under flips, rotations, or viewpoint changes, *BG-Er* remains stable (e.g., 12–17% for `ViT-L/14`). The findings imply that errors observed under these perturbations are more indicative of broad robustness deficits than spurious background correlations, underscoring the importance of disentangling these distinct failure modes.

**Model size and data:** Comparisons across architectures show that model size alone does not guarantee robustness. Large models such as `ViT-bigG` and `ViT-H/14(DFN-5B)` attain higher accuracy on the Bg-varied set (Table 3) yet exhibit substantial *BG-Er* under Scale (≈30–33%, Table 4).

In contrast, models trained on curated data (e.g., DataComp-1B) display reduced background reliance, indicating that pretraining data quality shapes shortcut behavior as much as model size.

**Backbone and resolution effects.** With fixed training data, architecture matters: within DataComp-1B, `ViT-B/16` records lower *BG-Er* than `ViT-B/32` across nearly all perturbations (30.5% vs. 50.7% under scale, Table 4), suggesting finer patching helps reduce background reliance. Scaling up models (e.g., `ViT-bigG`) achieve higher raw accuracy (Table 3) but still sustain background-driven errors (32.7% in scale, Table 4). Similarly, higher input resolutions (e.g., `ViT-B-SigLIP2` at `512px`) improve Bg-varied accuracy (64.9%, Table 3) but only marginally reduce *BG-Er* (39.3% under scale, Table 4), showing that resolution alone does not mitigate background reliance.

**Fine-grained confusion:** Factors like clutter, occlusion, and viewpoint shifts hinder discrimination between fine-grained classes (e.g., *chimpanzee* vs. *siamang*). As shown in Table 4 (*Fine-Er* columns), such errors persist across subsets, underscoring the inherent difficulty of the COVAR dataset. At smaller scales (last column, Table 4), models trade off *Fine-Er* against *BG-Er*. At larger patch sizes (e.g., `ViT-B-SigLIP2/32`), reduced access to detail further obscures fine distinctions and instead amplifies reliance on spurious background correlations.

**Takeaways.** Our analysis highlights concrete avenues for enhancing robustness in CLIP-like models. Scale emerges as the most challenging perturbation, affecting both accuracy and background reliance, underscoring the need to explicitly address scale variation during training. One approach is multi-scale feature alignment (Chen et al., 2025b), which fuses features across resolutions to reduce sensitivity to object size. Complementarily, RobustMixGen (Kim et al., 2025) augments data with diverse object–background combinations, enhancing robustness and mitigating reliance on spurious cues. Second, as viewpoint shifts degrade accuracy without increasing *BG-Er*, architectures that reduce over-reliance on 2D context, such as equivariant attention (Romero & Cordonnier, 2020), may help. Third, the persistence of high *BG-Er* in large models indicates that scaling alone cannot mitigate spurious background reliance. Curated pretraining data that decouples objects from typical contexts is crucial, as exemplified by ObjectNet (Barbu et al., 2019) and noise-robust training (Xiao et al., 2020). Together, these results underscore that data-centric strategies, rather than mere scaling, are essential for achieving improved robustness and generalization. Fourth, our backbone comparisons indicate that smaller patch sizes consistently mitigate background reliance. Hierarchical architectures such as the Swin Transformer (Liu et al., 2021), with finer-grained patch granularity, could support better object localization and reduced dependence on global context, making them a worthwhile direction for further investigation. Finally, persistent *Fine-Er* patterns indicate that robustness cannot be achieved by background debiasing alone. Prior work shows leveraging *fine-grained, category-specific textual descriptions* (Reed et al., 2016) and integrating *part-based localization* and *attention mechanisms* (Zheng et al., 2017; Fu et al., 2017) may help improve intra-class separability. Such strategies complement background robustness by helping models resist spurious correlations while capturing subtle distinctions needed for fine-grained categories. Overall, these findings point to the value of combining data curation, targeted augmentations, architectural refinements, and fine-grained supervision to advance model robustness.

# 6 SUMMARY

We considered the problem of CLIP's vulnerability to spurious correlations during model prediction and proposed a new visual interpretability technique, called CCI, for analyzing and understanding these issues. CCI generated attention maps by identifying semantically relevant groups of pixels and evaluating model changes with these regions masked out in the input, resulting in state-of-the-art performance on standard faithfulness benchmark datasets. By combining CCI with GroundedSAM, we showed that existing benchmarks, such as CounterAnimals, are insufficient for properly characterizing CLIP's error behavior. Our results showed that while these benchmarks attributed performance degradation primarily to background, the true underlying reasons point to other visual factors such as viewpoint variations, scale shifts, and fine-grained confusions. Building on this observation, we next proposed a new benchmark dataset, called COVAR, which systematically synthesized all these variations for each object category. We then used COVAR to benchmark the performance 18 different CLIP variants to both summarize the current state of their performance and provide insights and clear recommendations for how to improve these models.

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

# A APPENDIX

## A.1 ADDITIONAL QUALITATIVE CCI RESULTS

We present further qualitative comparisons of CCI against baseline methods in Figure 9 with a broader set of categories. Across diverse object types, CCI continues to generate heatmaps that are coherent, in contrast to the scattered or noisy activations produced by baselines. For instance, consider the *hair spray* in second row where CCI correctly attends to the spray bottles, while baselines either confuse or focus on small, disconnected fragments. Similarly, in challenging cases where object visibility is poor (see fourth row), CCI isolates the object of interest (*spider*) with sharper boundaries compared to baselines. These additional examples further provide strong evidence on the efficacy of proposed CCI method.

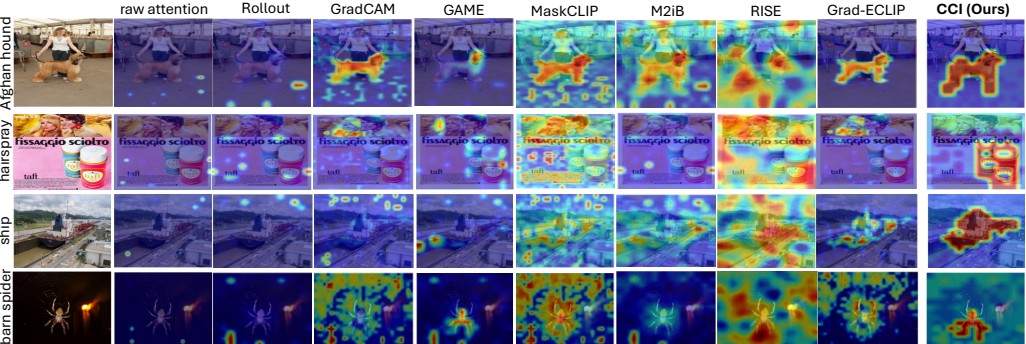

Figure 9: Additional Results comparing CCI against baseline interpretability methods.

## A.2 CCI RESULTS WITH OTHER CLIP VARIANTS

We repeat CCI computation with two other pretrained vision–language models, the OpenAI CLIP `ViT-B/32` at `224px` model (Radford et al., 2021) and a SigLIP variant (Zhai et al., 2023) `ViT-L/16` at `334px`, to evaluate the generality of our methodology. Figures 10 and 11 show results for the two variants repsectively. Findings are consistent with CLIP `ViT-B/16`: CCI generates concept-level, coherent heatmaps, indicating that CCI adjusts effectively to variations in backbone resolution and loss formulation. These findings demonstrate that CCI is independent of model and reliably generates comprehensible visual explanations for encoders from the OpenAI and OpenCLIP families.

## A.3 FOREGROUND MASK COMPUTATION

We leverage GroundedSAM (Ren et al., 2024) to generate foreground (FG) and background (BG) masks, which are then used to classify CLIP predictions as foreground-driven (*FG-Er*) or background-driven (*BG-Er*) across various datasets. For each image, we provide GroundedSAM with the prompt `<class name>, foreground objects`. The inclusion of "foreground objects" ensures that any distractor objects such as the *bucket* in Section 3.1 are correctly captured as part of the foreground mask. These masks then serve as a proxy for ground-truth object regions when computing Class-Conditional Importance (CCI) heatmaps. For each prediction, we compute the intersection-over-union (IoU) between the CCI heatmap and the FG/BG masks to determine whether the error is primarily foreground-driven (*FG-Er*) or background-driven (*BG-Er*).

We further validate GroundedSAM on the ImageNet-Segmentation (ImageNet-S) (Gao et al., 2022) validation set, which contains segmentation annotations on 12,419 images spanning 919 ImageNet categories. We create GroundedSAM masks using the same prompt and compare them against the dataset-provided masks. Figure 13 shows qualitative examples displaying the input image, GroundedSAM predicted mask, and the ground-truth mask. The average IoU between predicted masks and ImageNet-S masks is 0.93, demonstrating high alignment.

We show additional qualitative results on the CounterAnimals dataset in Figure 12. One can note that even in cases of heavy occlusion (see fifth row the figure), GroundedSAM correctly captures

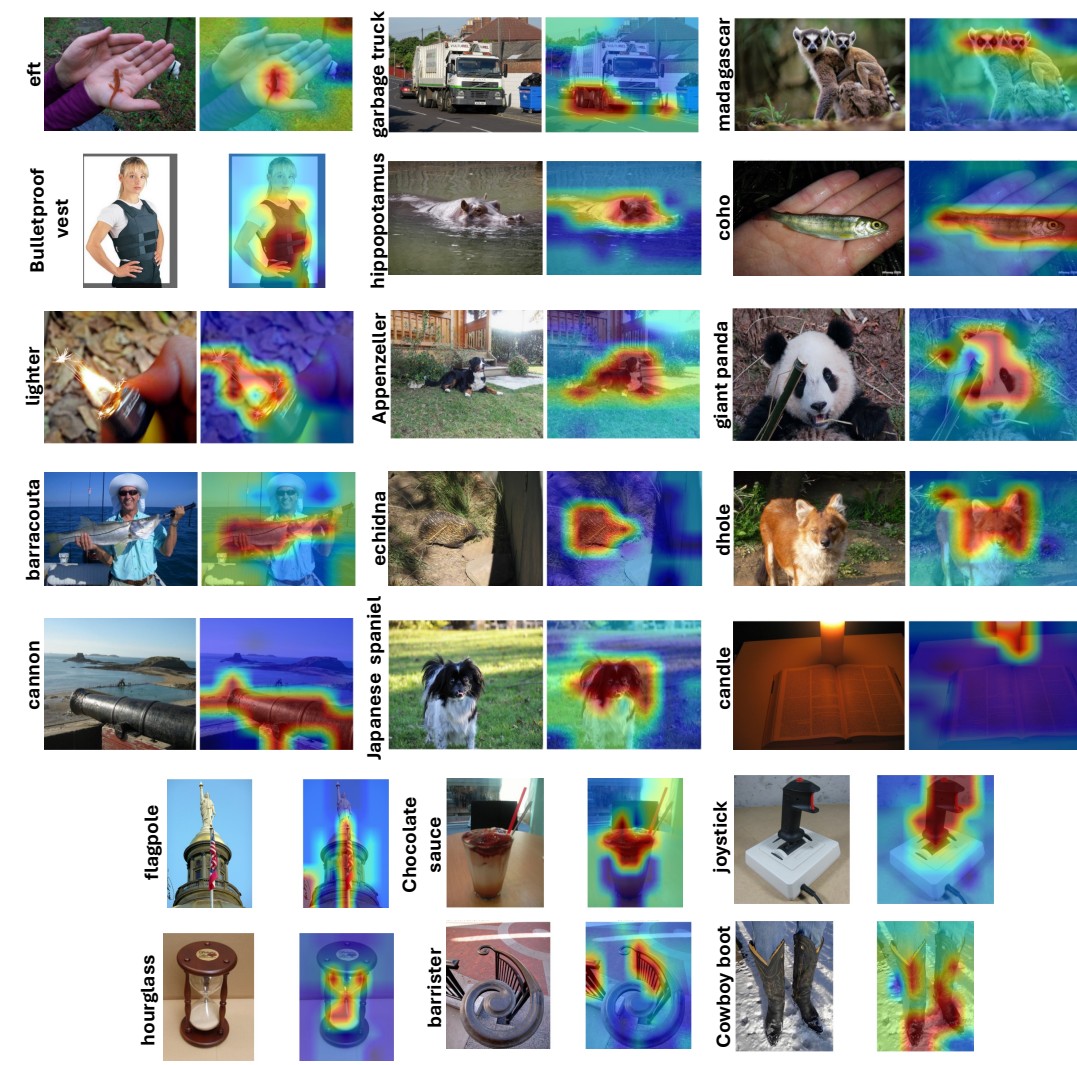

Figure 10: CCI Results with OpenAI CLIP `ViT-B/32-224px` model.

the foreground region of interest, illustrating that it can reliably captures foreground objects across diverse scenarios.

### A.4 PROMPTING DETAILS

#### A.4.1 FINEG COMPUTATION

As mentioned in Section 3.1 in the main paper, we used GPT-4o as a vision expert to determine whether the misclassified examples belonging to foreground-driven (*FG-Er*) represent fine-grained visual confusion or egregious failures. Below, we provide the exact prompt:

**Example Outputs:**

- Ground truth: *siamang*, Predicted: *chimpanzee* → `similar`
- Ground truth: *border collie*, Predicted: *australian shepherd* → `similar`
- Ground truth: *cat*, Predicted: *airplane* → `different`
- Ground truth: *lion*, Predicted: *bicycle* → `different`

**System Prompt:** You are a vision expert with deep knowledge of object categories and visual characteristics. Your task is to determine whether two categories are visually similar or clearly different based on appearance alone. Consider shape, texture, color, size, and typical visual features that a human would notice.
**User Prompt:** Ground truth class: [gt_class]
Predicted class: [pred_class]
Question: Evaluate whether these two categories are visually similar or clearly different. Consider the following:

1. Would a human observer easily confuse the two categories in a standard image?

2. Do they share key visual features (shape, color patterns, textures) that make them look alike?

3. If they are visually distinct and unlikely to be confused, classify them as different.

Respond with a single word only: similar if they are visually alike, different if they are clearly distinct.

By separating errors caused by mild intra-class similarity from more serious classification errors, this automated labelling gives us more information about the nature of model confusion than just accuracy metrics.

### A.4.2 CLASS SELECTION

As mentioned in Section 4.1, we curate a representative subset of classes from ImageNet while balancing semantic coverage and visual diversity. The goal is to avoid redundancy (e.g., multiple dog breeds) while still spanning a broad range of living and non-living concepts. To guide this process, we use GPT4o with the prompt shown below:

**System Prompt:** You are an expert in computer vision and dataset curation. Your task is to select a semantically and visually diverse subset of ImageNet classes for use in understanding spurious correlations in VLMs.
**User Prompt:** You are given a large set of 1,000 ImageNet classes. Your goal is to propose a smaller subset of about 30–40 classes that are semantically and visually diverse. Follow these guidelines:

1. Ensure coverage across living and non-living categories.

2. Avoid redundancy (e.g., do not include many dog breeds or many bird species). Select only a few representative ones.

Table 5 lists all selected classes included in our benchmark, providing the foundation for the subsequent background and structured variant generation.

### A.4.3 CURATING BACKGROUNDS

After selecting the classes, we systematically generate diverse background contexts for each image. Our aim is to disentangle model reliance on object appearance from contextual cues by creating multiple, semantically neutral backgrounds. The prompt used is below:

Table 6 lists all backgrounds included in our benchmark. These backgrounds cover natural, urban, and indoor environments, including water, snow, forest, desert, and indoor settings. Each background is applied to all 33 classes, creating systematic variants that allow us to disentangle object-specific recognition from spurious background reliance.

### A.5 CLASSWISE RESULTS WITH OTHER CLIP VARIANTS

To assess the generality of the background sensitivity patterns observed with CLIP ViT-B/16, we evaluated additional CLIP variants on the same 33,000 Bg-varied images. Our goal was to

Table 5: Final set of 33 classes selected for the benchmark, labeled with Class IDs.

| Class ID | Class Name | Class ID | Class Name |
|---|---|---|---|
| 1 | African elephant | 18 | park bench |
| 2 | Arabian camel | 19 | prairie chicken |
| 3 | Gila monster | 20 | pretzel |
| 4 | airship | 21 | rain barrel |
| 5 | alligator lizard | 22 | sea anemone |
| 6 | barn spider | 23 | slug |
| 7 | black swan | 24 | stove |
| 8 | bulbul | 25 | street sign |
| 9 | bullfrog | 26 | studio couch |
| 10 | cauliflower | 27 | submarine |
| 11 | chimpanzee | 28 | suspension bridge |
| 12 | dishwasher | 29 | trailer truck |
| 13 | electric locomotive | 30 | vulture |
| 14 | great white shark | 31 | warplane |
| 15 | hen | 32 | water ouzel |
| 16 | hermit crab | 33 | zebra |
| 17 | ice bear | | |

---

**System Prompt:** You are an expert in image editing and dataset creation. Your task is to propose diverse and realistic background settings for synthesizing objects in images.

**User Prompt:** Generate a list of 20 distinct background types that maximize diversity across scenes. The list should be independent of any specific object class and broadly applicable to placing different kinds of objects. Follow these guidelines:

1. Include both outdoor and indoor settings.

2. Ensure coverage across natural scenes, urban settings, and indoor environments.

3. Avoid repeating backgrounds that are too similar.

---

determine which aspects of the per-class accuracy drop are model-specific versus broadly expected across backbones.

Figures 15– 18 present classwise accuracy drops for various CLIP variants, with plots labeled from (a) to (m). For example, in plot (a) (class ID 6), the OpenAI CLIP `ViT-B/32` model exhibits a substantially higher relative accuracy drop compared to the same class under CLIP `ViT-B/16` (80 vs 20%). On the other hand, many other classes, such as IDs 1 and 33, continue to show smaller drops, consistent with the pattern seen in `ViT-B/16`.

Overall, while some classes behave differently across variants, the broader patterns are consistent: strongly background-dependent classes continue to exhibit significant drops, and those that were resilient to background variation with `ViT-B/16` typically remain resilient with other backbones. This highlights the usefulness of our benchmark for examining spurious correlations and implies that the observed background effects are primarily dataset- and class-intrinsic rather than an artifact of a particular model.

## A.6 IMPLEMENTATION DETAILS OF VARIANTS

Here we provide implementation details for the variants generated in our proposed COVAR dataset.

**Background variants.** We use Emu2 (Sun et al., 2024) to create background variants. For each class, we kept the original object in the foreground while changing the background to various natural settings. The model received instructions like *"background description. Edit only the*

Table 6: Curated backgrounds for the benchmark. Each background is applied to all 33 classes to generate systematic variants.

| Bg ID | Description |
|---|---|
| 1 | railway track in an outdoor setting |
| 2 | colorful garden with flowers and greenery |
| 3 | dense tropical forest |
| 4 | farmyard |
| 5 | hot desert with sand dunes |
| 6 | open grassland with tall green grasses |
| 7 | swampy area |
| 8 | cozy living room |
| 9 | savanna |
| 10 | calm ocean with clear blue water |
| 11 | fluffy white cloud in a bright blue sky |
| 12 | highway with empty road stretching behind |
| 13 | rocky shore with waves |
| 14 | rocky terrain |
| 15 | snowy landscape |
| 16 | beach |
| 17 | forest floor with leaves and sunlight filtering through trees |
| 18 | tree branches in a leafy forest |
| 19 | crowded marketplace with people and stalls |
| 20 | night cityscape with artificial lights |

*background and keep the foreground subject intact"*. This ensured that only background pixels were changed, while the object's identity and position remained the same.

**Viewpoint variants.** To achieve viewpoint diversity, we used Zero123+ (Shi et al., 2023), a text-to-3D image synthesis method. For a given original image, Zero123+ generates 6 new viewpoints of which we randomly select 2. We created images using 75 inference steps, which were enough to keep fine details in general objects while ensuring consistent viewpoint changes.

**Scale variants.** We produced scale variants with Stable Diffusion inpainting (Rombach et al., 2022) by outpainting the original image onto larger canvases. Each image was expanded to different scale factors (up to $8\times$), keeping the original object centered (see Figure 19 for an example). The inpainting mask (generated using GroundedSAM as described in A.3) made sure that only the surrounding areas were generated, preserving the original foreground content. We used a standard classifier-free guidance scale of 7.5 and applied 30 diffusion steps for all images.

**Others.** We applied five standard geometric transformations using OpenCV, designed to preserve semantic content while perturbing pixel-level statistics:

- **Rotation:** images were rotated by a random angle in $[-45°, 45°]$, with borders filled via Stable Diffusion inpainting.

- **Horizontal and Vertical Flips:** standard left–right and top–bottom flips.

- **Crop:** cropping a region from center covering 60–90% of the original image area, followed by resizing.

- **Translation:** shifting the image up to 20% of its width/height in each direction, with borders filled via Stable Diffusion inpainting.

Together, these methods produce a well-rounded set of variants that enable controlled evaluation of background reliance, viewpoint generalization, scale sensitivity, and standard geometric robustness.

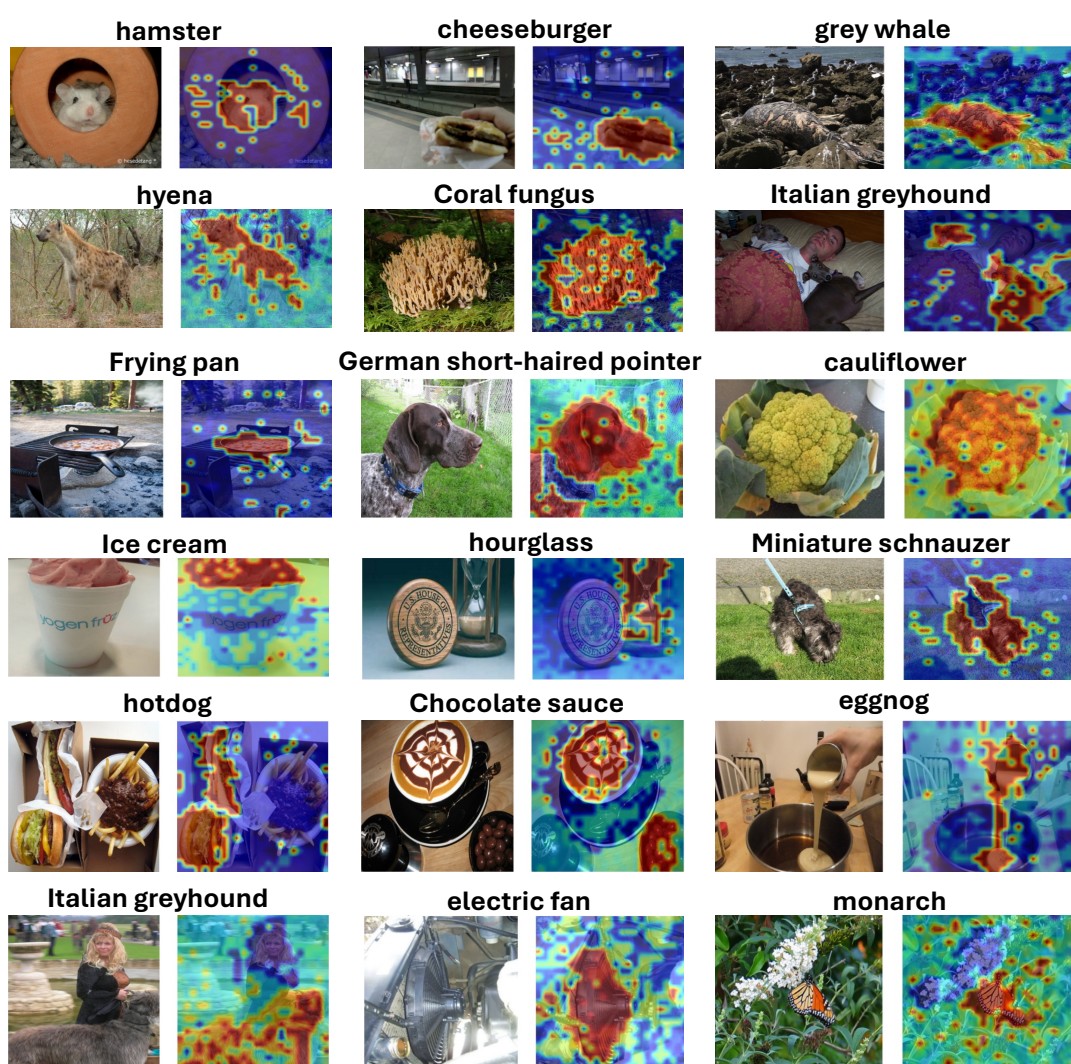

Figure 11: CCI Results with SigLIP `ViT-L/16-334px` model.

## A.7 CLASSWISE RESULTS FOR ALL SUBSETS

Figure 20 presents per-class accuracy drops across seven subsets (excluding the bg-varied subset, which is shown in the main paper in Section 4.2). For each class, we compute the accuracy over all images in a subset and compare it to the original ImageNet accuracy for that class.

The plots in Figure 20 show that for all subsets other than scale, accuracy degradation varies significantly across classes: some classes remain consistently robust, while others show notable sensitivity. In contrast, for the scale subset in plot (a), every class exhibits a substantial drop, with a minimum decline of approximately 25%. For example, class 33 shows a modest average drop of 5% across the non-scale subsets, but under the scale subset, the drop is almost 30%, illustrating that scale affects this class much more strongly than the others. Overall, these results indicate that CLIP's robustness is both class- and subset-dependent, with scale having a uniformly strong impact across all classes.

## A.8 CCI RESULTS ON COVAR

Figure 21 presents CCI visualizations on samples from COVAR, illustrating how CLIP's focus shifts under different variant conditions. In the first row, we show a pair of images where the right image

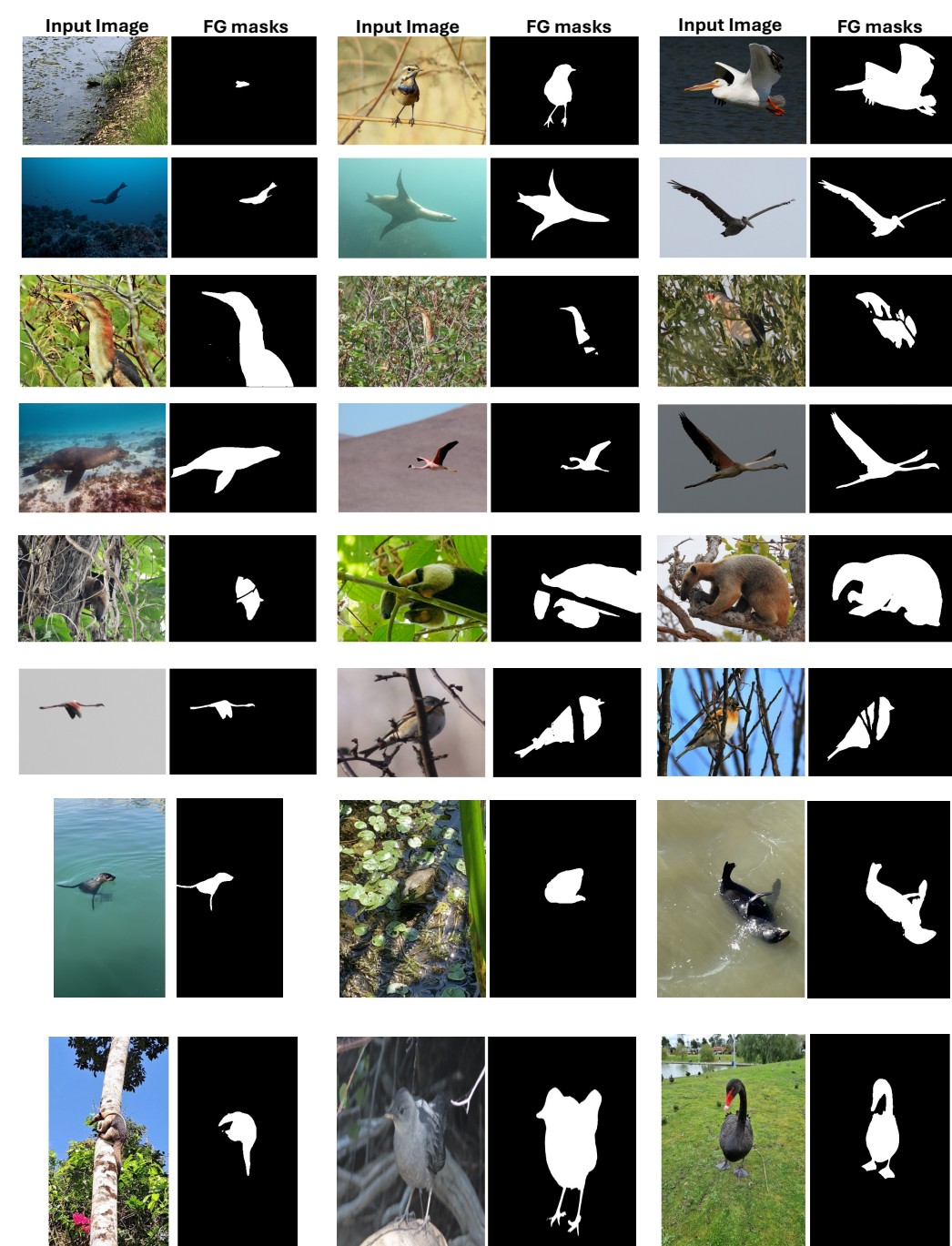

Figure 12: Qualitative results of GroundedSAM on the CounterAnimals dataset.

is a scale-reduced variant of the left. On the original image, CCI correctly focuses on the foreground and predicts the class as *African elephant*. However, in the scale-reduced variant, the model's attention shifts toward the background resulting in a misprediction as a *freight car*, likely due to the railway-track context. The second row depicts a *barn spider* in two different backgrounds: while the model accurately predicts the left image, it attends to the beach background in the right image, erroneously predicting a *crab*. Subsequent rows illustrate additional qualitative patterns, such as v-flip variants where predictions are incorrect yet the model still focuses on the foreground. Notably,

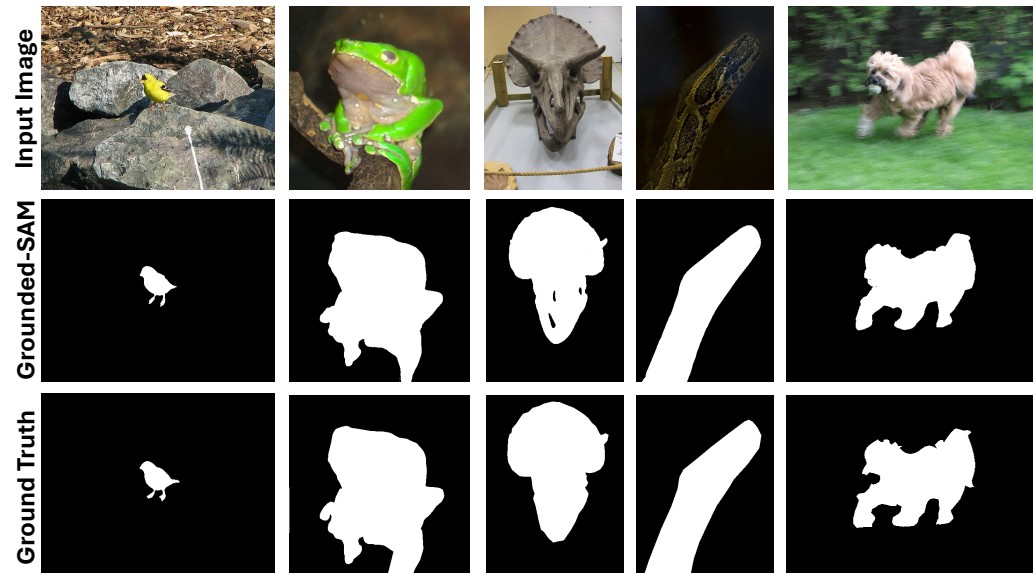

Figure 13: Comparison of GroundedSAM predicted foreground masks with the ImageNet-S ground-truth segmentation masks. For each image, the first row shows the input image, the second row shows the GroundedSAM predicted mask, and the last row shows the ImageNet-S ground-truth mask.

in such cases, the misclassified classes remain visually similar to the ground truth, for example predicting a *moving van* instead of a *trailer truck*. These examples complement the main paper's insights, demonstrating that even when accuracy drops for certain variants like v-flip, the fraction of background-driven correlations remains largely unchanged.

## A.9 AGGREGATING PREDICTIONS BY BACKGROUND CONTEXT

To complement the quantitative analysis of robustness and spurious correlations, we also conducted a qualitative examination of CLIP's background biases by aggregating its predictions across images sharing the same background context. Specifically, for each background type, we averaged predictions over all corresponding images and recorded the most frequently predicted classes. Table A.9 summarizes representative results for a subset of backgrounds. This view reveals strong, dataset-wide associations between certain backgrounds and particular object categories- for example, railway tracks strongly elicit predictions of *locomotive* or *bullet train*, even when the ground-truth object is unrelated. Such correlations are likely a reflection of CLIP's training distribution, where railway tracks frequently co-occur with trains, leading the model to overweight background context as a cue for object recognition. These observations highlight that CLIP's predictions are often guided more by contextual cues than by the objects themselves, underscoring the importance of explicitly disentangling object and background information in evaluating model behavior.

Table 7: Top Predicted Classes per Background (Averaged across all images)

| Background | Classes |
|---|---|
| **railway track** | locomotive, bullet train, freight car |
| **rocky shore** | water ouzel, hermit crab |
| **garden** | rain barrel, park bench |
| **tropical forest** | chimpanzee, bulbul |
| **sky** | vulture, airship |
| **road** | zebra, trailer truck, street sign |
| **swampy area** | bullfrog |
| **desert** | Arabian Camel |

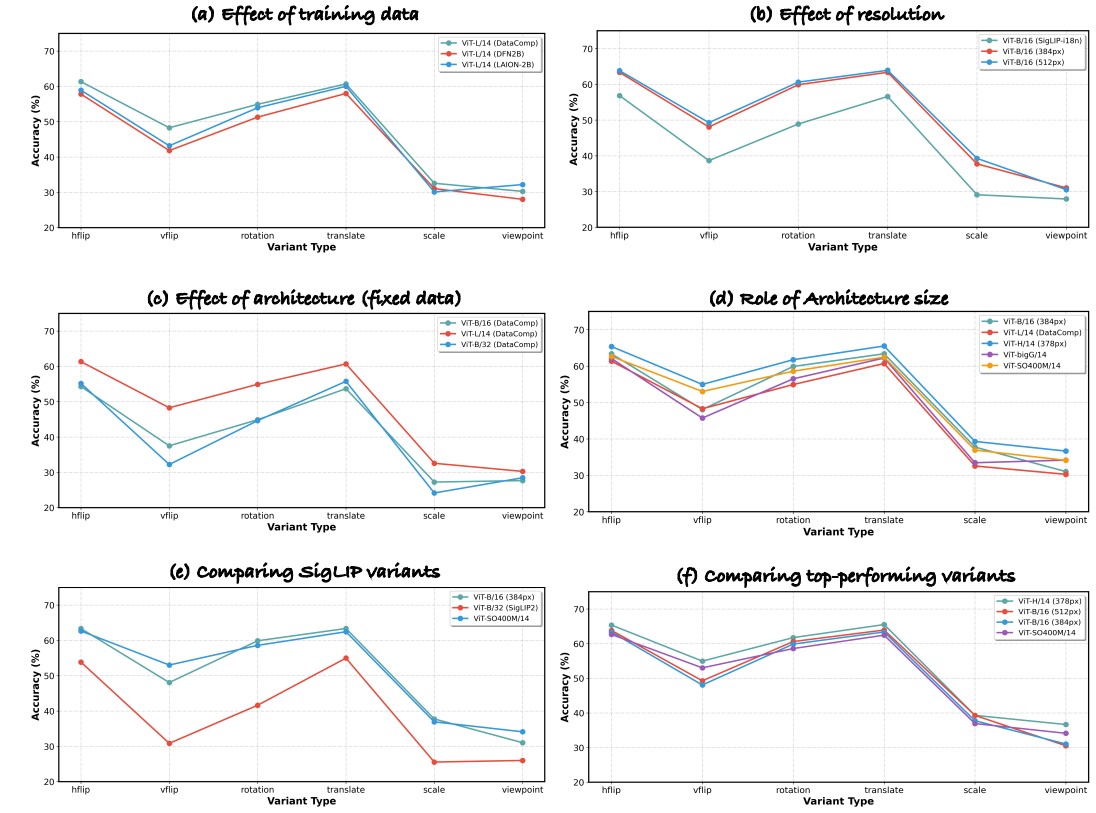

Figure 14: Average CLIP accuracies across dataset subsets.

## A.10 LLM USAGE

We utilised LLM for retrieval and discovery of relevant papers in the literature, and for rephrasing/rewording few paragraphs. This paper's research concepts, methodology, analysis, and insights are all unique and LLM has absolutely no role to play in that process.

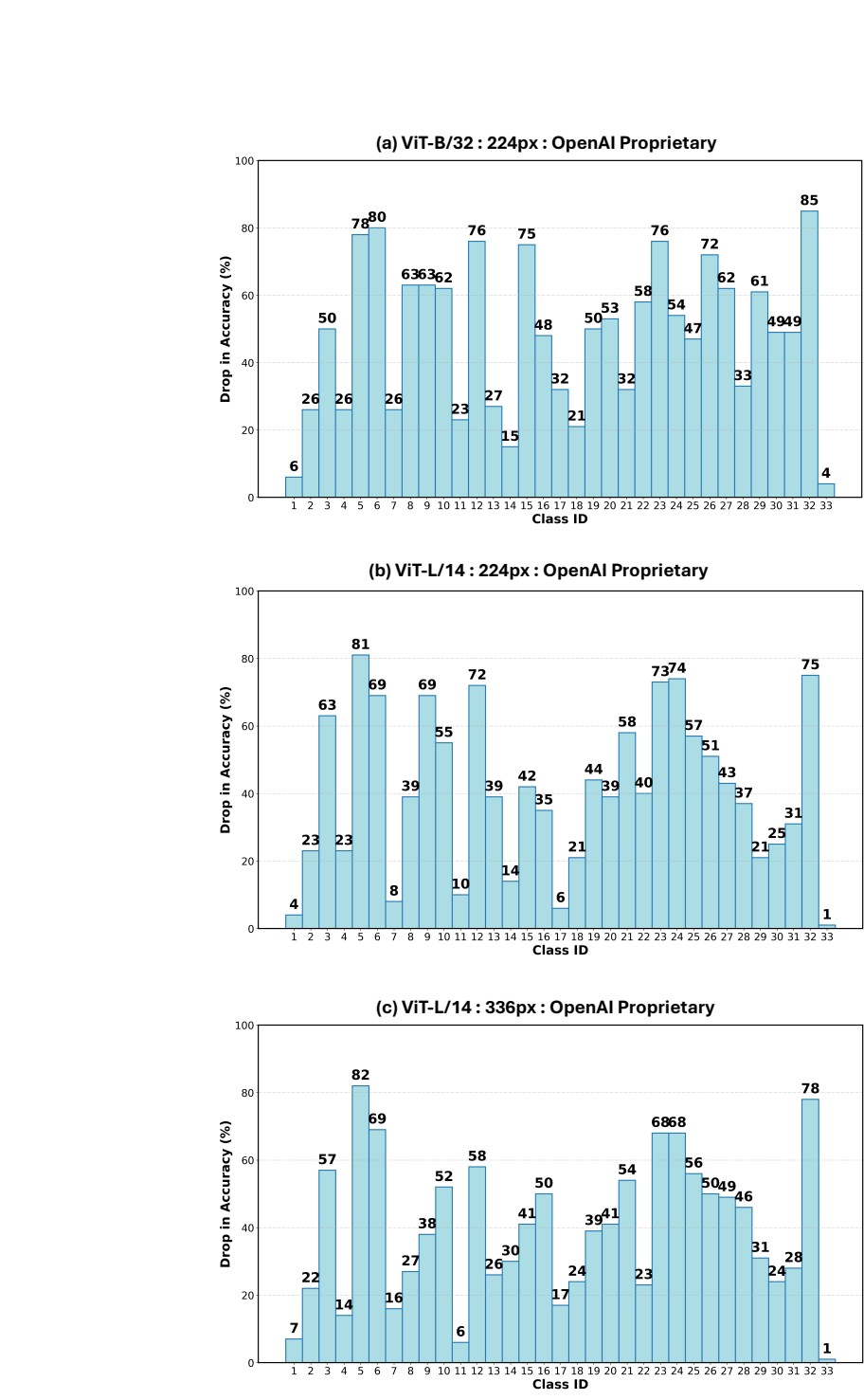

Figure 15: Classwise accuracy drops across OpenAI CLIP variants.

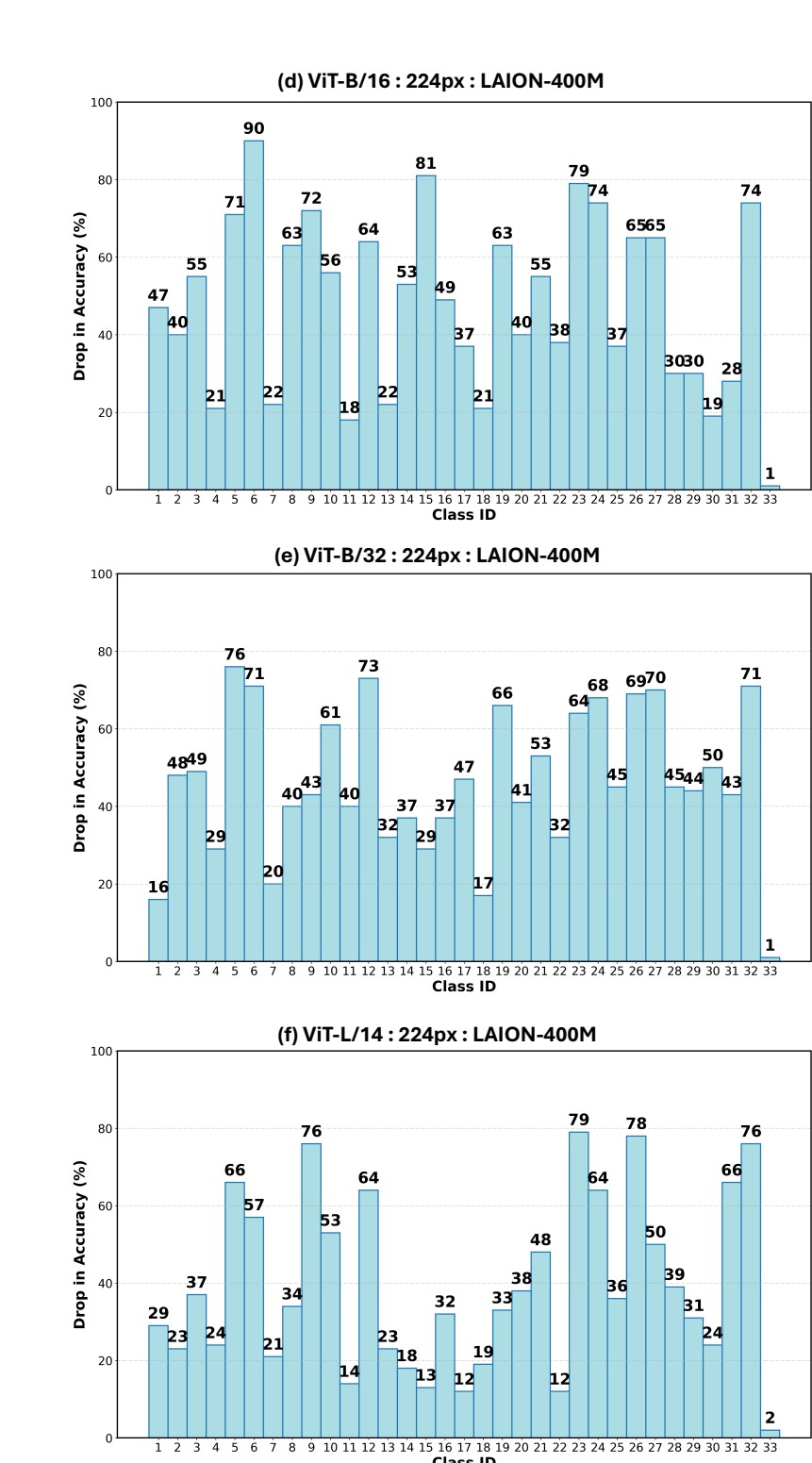

Figure 16: Classswise accuracy drops across OpenCLIP variants pretrained on LAION-400M.

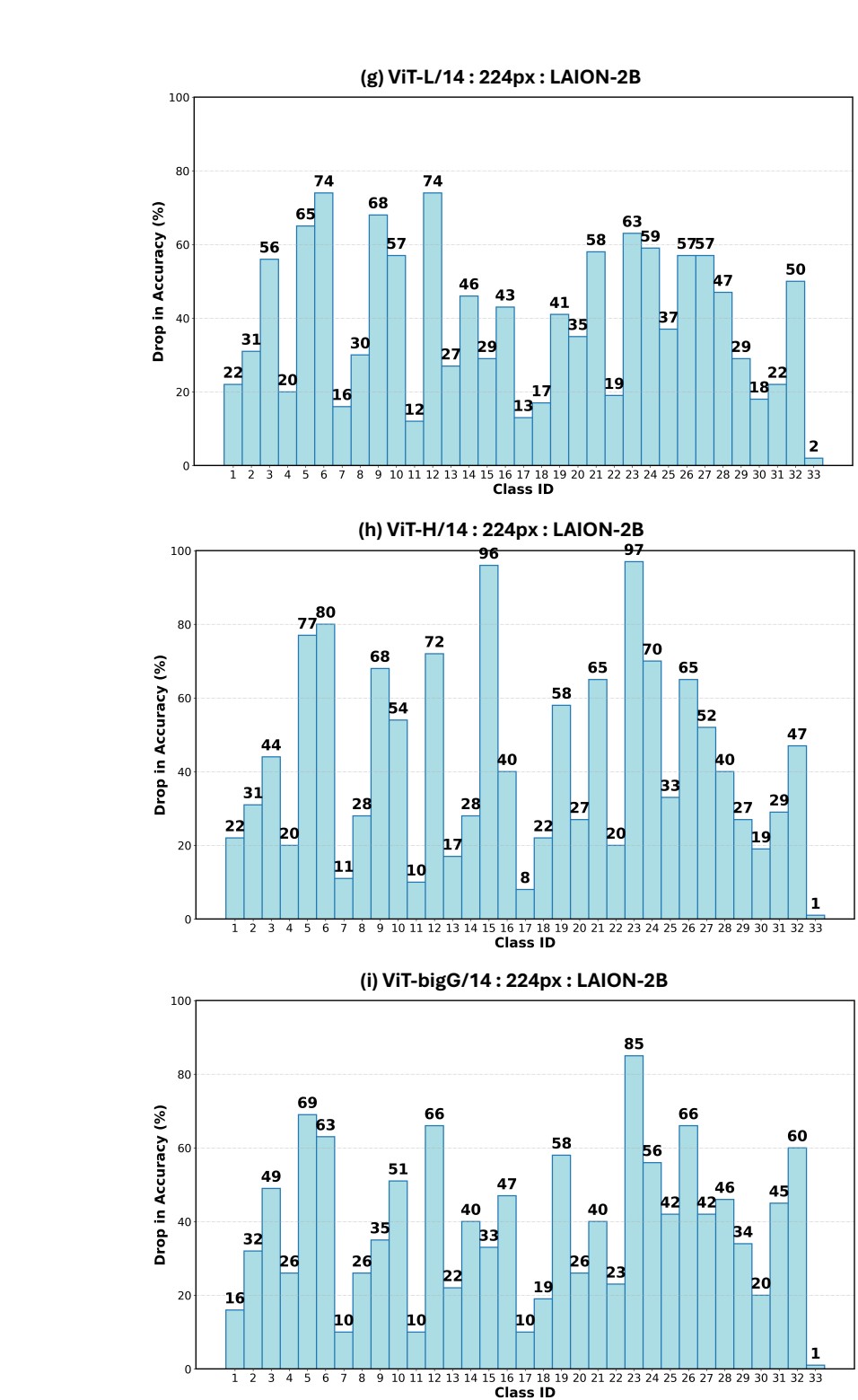

Figure 17: Classwise accuracy drops across OpenCLIP variants pretrained on LAION-2B.

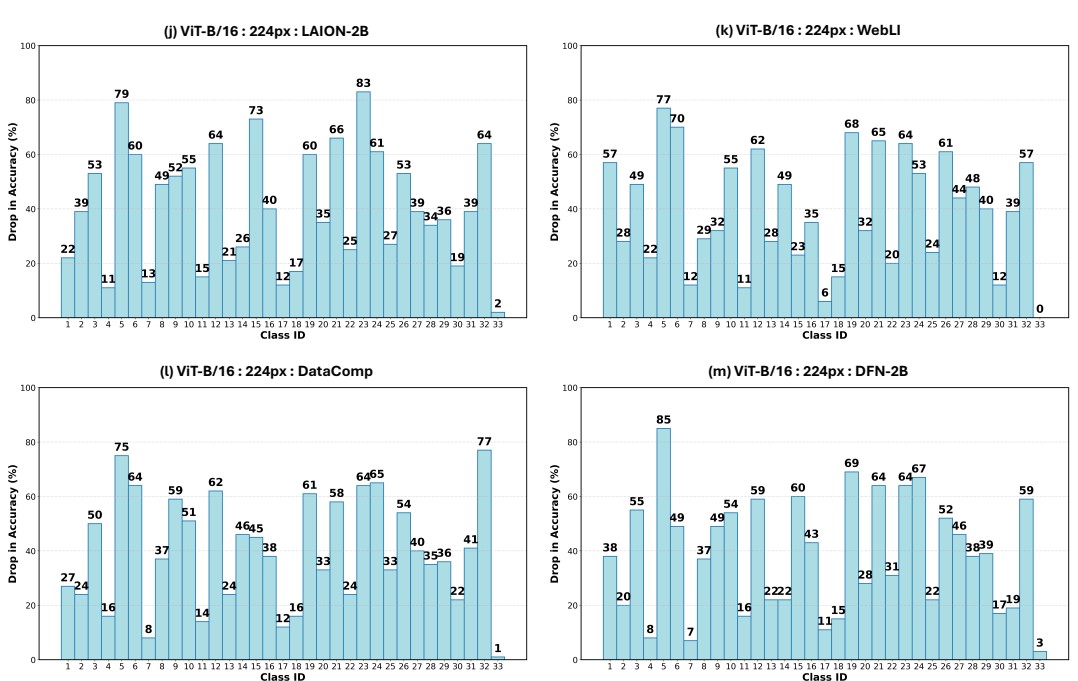

Figure 18: Classwise accuracy drops across `ViT-B/16` OpenCLIP variants on different datasets.

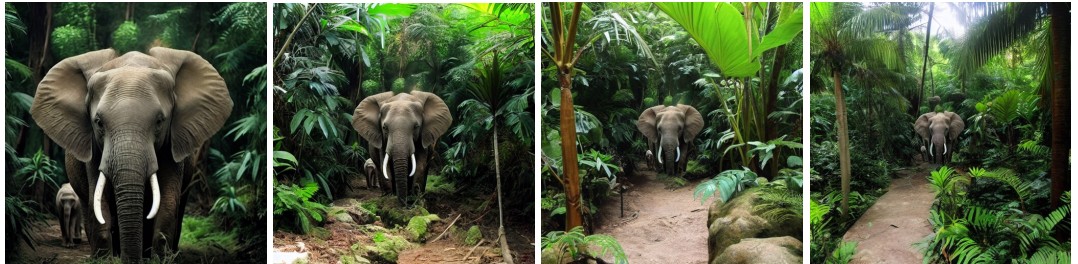

Figure 19: Example demonstrating varying scales for the same input image.

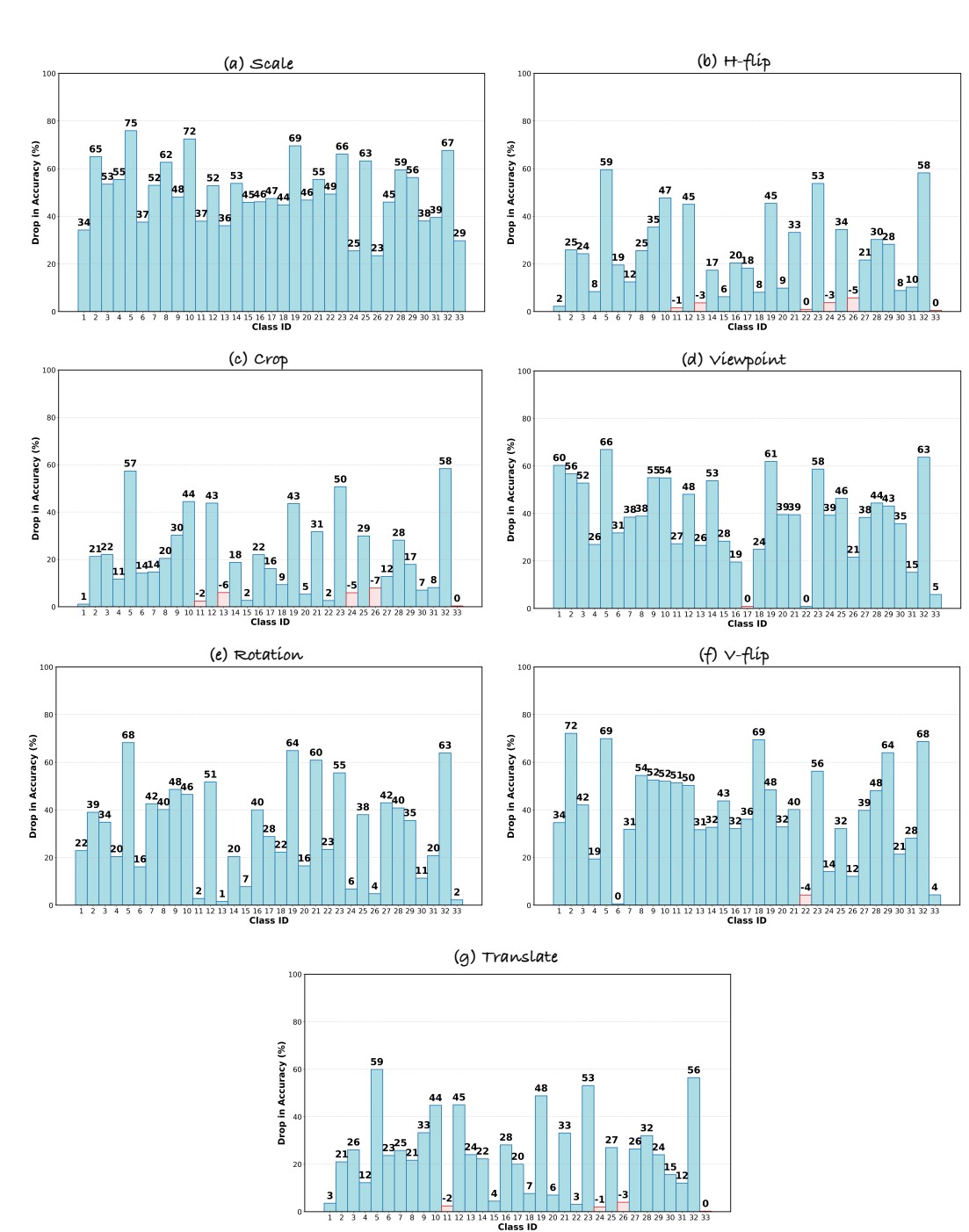

Figure 20: Per-class accuracy drops across various dataset subsets.

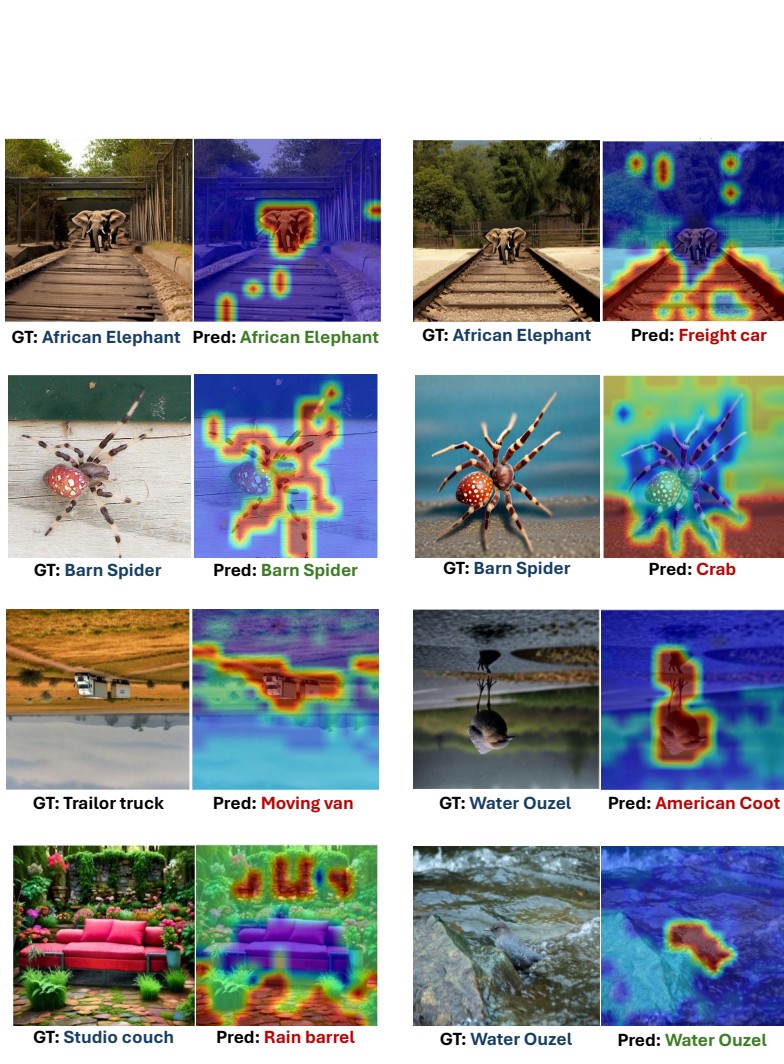

Figure 21: Qualitative CCI Results on COVAR.

