# OpenReview forum: "Foreground or Background? Visual Interpretability and Robustness Analysis of CLIP"
_ICLR.cc/2026/Conference — ICLR 2026 Conference Withdrawn Submission_

### Official Review · Reviewer_ZFmJ · 2025-10-29

**Soundness:** 2
**Presentation:** 2
**Contribution:** 2
**Rating:** 2
**Confidence:** 4

**Summary:**

The paper presents an interpretability method for CLIP. It first groups image tokens into clusters, and then masks them and finds the effect of each cluster on the similarity to the text embedding. The method is evaluated via deletion/insertion AUC over retrieval on MS-COCO, and is shown to outperform various gradient based interpretability methods. It also presents a benchmark to evaluate models on other distribution shifts such as rotation, cropping, translation, zooming in/out, etc.

**Strengths:**

1. The proposed method outperforms many gradient based interpretability methods on the deletion AUC metrics
2. Paper is well written

**Weaknesses:**

1. The novelty of the method is severely limited. The masking method is identical to Jain et al [1] but not cited in the paper. Clustering features and scoring is also not new in interpretability, methods like LIME and SHAP are designed to assign importance scores in this manner (see also super clustering methods in traditional computer vision which create cluster in pixel space, for example SLIC). In fact, Jain et al combine their masking method with LIME and show improvements on similar metrics.

2. The benchmark is also not very novel, distribution shifts such as rotation, cropping, translation are very common and thus implemented in PyTorch to provide synthetic data augmentations. Distribution shifts related to background are also well studied with numerous benchmarks, see Waterbirds [2], ImageNet-9 [3], D3S [4], etc.

3. The interpretability method and benchmark do not seem to have any relationship, for example CCI is not used on the new benchmark. It would be better to split the paper into two.



[1]  Jain, S., Salman, H., Wong, E., Zhang, P., Vineet, V., Vemprala, S.H., & Ma̧dry, A. (2022). Missingness Bias in Model Debugging. ArXiv, abs/2204.08945.

[2] Sagawa, S., Koh, P., Hashimoto, T.B., & Liang, P. (2019). Distributionally Robust Neural Networks for Group Shifts: On the Importance of Regularization for Worst-Case Generalization. ArXiv, abs/1911.08731.

[3] Xiao, K.Y., Engstrom, L., Ilyas, A., & Ma̧dry, A. (2020). Noise or Signal: The Role of Image Backgrounds in Object Recognition. ArXiv, abs/2006.09994.

[4] Kattakinda, P., Levine, A., & Feizi, S. (2022). Invariant Learning via Diffusion Dreamed Distribution Shifts. ArXiv, abs/2211.10370.

**Questions:**

Please address the weaknesses listed above

---

### Official Review · Reviewer_RxEy · 2025-10-30

**Soundness:** 2
**Presentation:** 2
**Contribution:** 2
**Rating:** 4
**Confidence:** 3

**Summary:**

The paper investigates CLIP’s reliance on background context and its limited robustness to visual variations. It proposes CCI, an interpretability approach that identifies how different image regions influence CLIP’s image-text matching, and introduces a benchmark that systematically alters object and background features.

**Strengths:**

1- The paper presents CCI, a clear and practical training-free interpretability method that highlights how different image regions influence text-image similarity.

2- The paper introduces COVAR, a well-constructed benchmark that effectively separates object and background elements to enable controlled robustness testing.

**Weaknesses:**

1- The paper does not clearly explain what its main contribution is or how it meaningfully differs from previous work on interpretability and background/object bias in vision-language models.

2- The proposed method has been evaluated only on a set of CLIP variant models, and its generalizability to other vision-language models remains unclear.

3- The paper does not discuss the limitations of the proposed method or analyze its failure cases.

4- The paper includes some writing and formatting issues, for example, equations are not numbered, and in the formula on line 181, the correct notation should be written as A^l_k(i, j). Additionally, Tables 3 and 4 are not easily readable and would benefit from clearer formatting or layout adjustments.

**Questions:**

1- Could you clarify what distinguishes this work from prior studies on interpretability and background/object bias in vision-language models?

2- Could you evaluate the method on other vision-language models beyond CLIP to better show its generalizability across different architectures?

3- Instead of relying solely on cosine similarity between text and image embeddings to estimate cluster importance, could a more reliable or semantically grounded metric be used?

4- How sensitive is the CCI method to the choice of clustering algorithm and its hyperparameters?

---

### Official Review · Reviewer_MuEZ · 2025-10-31

**Soundness:** 2
**Presentation:** 2
**Contribution:** 3
**Rating:** 2
**Confidence:** 5

**Summary:**

The goal of the work is to highlight that "correct predictions may rely on background cues while errors can occur despite object-focused attention."

- CCI is a region-level interpretability method that identifies meaningful regions for model prediction/accuracy.
- CCI identifies the image regions via image–text similarity scores, grouping patches into clusters. Masking these patches helps measure changes in model predictions relative to the unmasked input, highlighting the significance of the masked patches.
 - CCI analysis disentangles background and foreground contributions, helping visualize/analyze reliance of CLIP's predictions on various aspects of images.

- The COVAR dataset evaluates models' reliance on spurious correlations, like background and foreground.

**Strengths:**

- Table 1, 2 & Fig 4 shows that CCI "highly weighted patches" have better recovery in accuracy when introducing them, while an adverse drop in accuracy when deleting those patches.

- CCI is very easy to implement and has an intuitive explanation. Figure 3 shows good qualitative proof for the method.

- Paper highlights the structural limitations of CA dataset, indicating binary easy–hard split overlooks variations in viewpoint, scale, pose, and composition.

- Unlike the previous work of Coarse partitioning, the COVAR dataset introduces fine-grained partitions (splits) to further expose models' vulnerability on various augmentations.

**Weaknesses:**

- **Paper is not well written**:
  - Line 206: the slug image (row 1), what row, what's the figure number?
  - What are all the classes 11, 22, 31, 5, 23, 32?
  - The introduction paints a vague picture of what this paper is trying to do. CCI is trying to highlight areas that models focus on during prediction.
  - Generic statements assume reader familiarity with explainable AI / Interpretable AI, while not being standard computer vision terms: i) Line 040 “spurious correlations” ii) Line 036 “that is, associations driven by dataset biases rather than true semantic grounding” iii) Line 039 “Such correlations reduce robustness under distribution shifts”: what distribution shift? Noise? Domain? iv) Line 045 “probe sensitivity to atypical backgrounds” v) Line 052: “although 32.1% of the easy set and 46.6% of the hard set are misclassified, the proportion attributable to background reliance remains nearly identical (6.23% vs. 7.26%, BG-Er, Figure 1(d))”  vi) Line 90 (i) accuracy is an inadequate proxy, e.g., conflating background reliance with fine-grained class confusions,





- **Figure 1 caption is vague and non-informative**: Figure 1 doesn’t convey what we are looking at. (a) Is that grad cam?, What do we infer from attention? (b) What's CA? “[water]” is a prediction?  (c) What's COVAR? Is this your dataset? (d) What exactly is the y-axis? How was the root cause of the error “BG-Er” determined? Line 096: (e.g., Figure 1, first row). This is especially problematic as the description of this figure is given on Line 300 on page 6, and Line 323 on Page 6. Sub-figure captions' fonts are difficult to read.

- What makes the CCI “CLIP” centric? Design Choices / ablation on CCI missing
    - What about other VLMs, like EVA, BLIP, CoCa, SigLIP, and ALIBEF?
    - What about Pyramid Transformers like Swin, Twin?
    - If one masks such tokens in window-based self-attention, spatial coherence might be broken.
    - What happens if you just set patches to zero instead of modifying self-attention?
    - How good are background/foreground masks? Since they themselves were generated via another model (GroundedSAM), and if they are making predictions based on backgrounds to mask?
    - How good are masks on COVAR dataset? if they are generated via some model as well, and they struggle with transformation?
    - What happens with different values of k clusters in CCI?
    - What about other spatial tokens across layers in ViT? Why only use Patch embedding?

- **Is Table 1, 2, Figure 4 really meaningful?** The scores are computed based on “which patches drop the performance the most”. Then the CCI is evaluated on “How dropping/inserting high-scoring patches affects performance”. Additionally, how the scores are computed is a bit unclear. Heatmaps are produced to correct labels (line 188)? So even if the model predicted wrong, the heatmap corresponding to the correct class was used to evaluate Table 1, 2, and Fig. 4?

- **Problematic Claim Line 333** *“Some categories are robust to background variation while others are strongly background-dependent”*. There is another explanation for this phenomenon. In adversarial learning, we know slight tweaks in pixel values let the model’s prediction go haywire. Panda --> airplane via adversarial noise.  This drop in accuracy may be more caused by how “Emu2 image editing” is generating these synthetic images/backgrounds for certain classes. Similar explanation for “two different viewpoints,” where synthetic noise and background noise can influence the model prediction. Cropping often improves it; further supporting this, removing the noisy background improves performance. Are the images sharing the same background, or does each image get its own background?

- **Propose a solution?** The COVAR dataset exposes models' vulnerability on foreground and background. Is there a solution, on how to improve performance?

- Paper is a joint of two different topics :
   - **CCI new kind of receptive field visualization (analysis)**. The analysis is really weak without maximizing potential on exploring interesting insights of what models focus on while making predictions. Current analysis is limited to lines 294 - 304.
   - Redundant benchmark **COVAR benchmark (noises, like scale, background)**: This is a very well-discussed topic. Scaling is a very likely pixelation, a very well-researched topic. Similarly, changing background is a fairly discussed topic.
  - Overall, the paper is very weak on novelty. CCI weak analysis with redundant COVAR benchmark.

**Questions:**

- Line 180 (equation number missing) is A[i,j] same as M[j]? I think these “j”s are different. One is the jth column while other is “j” token in flattened patches. Or line 176 m(j) is missing “i” i.e. m(i,j).

- Grad-ECLIP is doing reasonably good as well. Figure 10: corresponding predictions would really be appreciated.

- Can the authors please highlight what novelties (method/insight/analysis) are for this work? Novelties, as in inferences that have not been published before.

---

### Official Review · Reviewer_B8Q5 · 2025-11-01

**Soundness:** 2
**Presentation:** 3
**Contribution:** 3
**Rating:** 6
**Confidence:** 5

**Summary:**

This paper addresses the vulnerability of CLIP models to spurious correlations during classification and introduces a new interpretability method called Cluster-based Concept Importance (CCI), which clusters image patches based on their similarity. The authors also propose a dataset named Controlled Variants (COVAR), designed to overcome the limitations of previous benchmarks by placing objects in diverse backgrounds and systematically varying visual factors. Together, these contributions aim to provide a deeper understanding of CLIP’s decision mechanisms and its susceptibility to background-related biases.

**Strengths:**

- The COVAR dataset introduced by the authors can significantly advance research on spurious correlations and distributional robustness. It enables researchers to systematically evaluate model robustness under various augmentations (distributional robustness) and background variations (spurious correlations).

- Compared to existing datasets such as Colored MNIST (CMNIST), which is overly simplistic, or Waterbirds, which lacks sufficient diversity, COVAR provides a more complex and diverse benchmark that is better suited for modern robustness studies.

- Experiments are conducted on both general interpretability methods and CLIP-specific explanation techniques, demonstrating the versatility of the proposed approach.

- The comprehensive evaluation across multiple ViT model variants effectively illustrates the generalizability of the method.

**Weaknesses:**

- The authors should report the actual spurious correlations between foreground and background classes, along with additional statistical details about the dataset. Although the COVAR dataset could be of great value to the community, its unavailability prevents a thorough assessment of its quality.

- The reported accuracy drops are influenced not only by the COVAR dataset but also by the training data used. Providing a correlation matrix or similar analysis of spurious relationships would improve interpretability of the results.

- Figure 6 shows that while most classes experience uneven but often substantial drops in accuracy, a few retain performance or even improve. Specifically, the ice bear class (ID 17) shows a 10% increase in accuracy. The authors should explain this phenomenon. More generally, the influence of background selection on Figure 6 and the corresponding raw accuracies should be discussed.

- It is unclear whether the results were obtained using a single random seed. The authors should report results averaged over multiple seeds to account for potential variance.

- The procedure for selecting K (the number of clusters) and the sensitivity of CCI to different K values should be clarified and analyzed.

- The code has not been released, which limits reproducibility and verification of the reported results.

Minor Issues:
- Grouping the classes in the COVAR dataset based on their semantic similarity (e.g., animals) or by the degree of spurious correlation they exhibit could help clarify the observed variations in class-wise results.

- The test data are not i.i.d due to correlations between images generated from different visual transforms, and this should be explicitly acknowledged and considered in the analysis.

**Questions:**

Please address my concerns raised in the Weakness section.

---

### Note · Authors · 2025-11-14

I have read and agree with the venue's withdrawal policy on behalf of myself and my co-authors.